

# Electron currents from gradual heating in tilted Dirac cone materials

Ahmadreza Moradpouri, Mahdi Torabian and Seyed Akbar Jafari⋆

Department of Physics, Sharif University of Technology, Tehran 11155-9161, Iran

⋆ jafari@sharif.edu

## Abstract

Materials hosting tilted Dirac/Weyl fermions provide an emergent spacetime structure for the solid state physics. They admit a geometric description in terms of an effective spacetime metric. Using this metric that is rooted in the long-distance behavior of the underlying lattice, we formulate the hydrodynamic theory for tilted Dirac/Weyl materials in $2 + 1$ spacetime dimensions. We find that the mingling of space and time through the off-diagonal components of the metric gives rise to: (i) heat and electric currents proportional to the *temporal* gradient of temperature, $\partial_t T$ and (ii) a non-zero Hall-like conductance $\sigma^{ij} \propto \zeta^i \zeta^j$ where $\zeta^j$ parameterize the tilt in $j$'th space direction. The finding (i) above that can be demonstrated in the laboratory, implies that the non-trivial emergent spacetime geometry in these materials empowers them with a fascinating capability to harness naturally available sources of $\partial_t T$ of hot deserts to produce electric current. We further find a tilt-induced non-Drude contribution to conductivity which can be experimentally disentangled from the usual Drude pole.

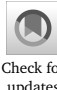

## 1  Introduction

The motion of electrons in conductors in the absence of external temperature and/or electro-chemical gradients is purely random thermal motion [1]. Once a spatial temperature gradient $\nabla T$ is introduced, the vector character of $\nabla$ specifies a preferred direction in the space, and therefore electrons flow along $\nabla T$ [2]. The purpose of this paper is to propose a class of materials where a temporal gradient $\partial_t T$ *alone* (i.e. without requiring any spatial gradient) is sufficient to generate electric and/or heat currents.

To set the stage, suppose that we have an anisotropic material where there is a preferred direction in the space determined by a vector $\zeta$. Such a preferred direction is expected to somehow organize the random motion of electrons by preferring the orientation set by the vector $\zeta$. In this paper, we consider a class of anisotropies that can be encoded into a spacetime metric, and the favored direction $\zeta$ determines the entries of the metric that mix space and time. Such a metric influences the motion of electrons in such a way that space and time coordinates of the motions mingle. In this way it is natural to expect that a pure temporal gradient of a spatially uniform temperature $T$ field to drive a current.

How do the new spacetime structures arise in the solid state? The electrons in the solids are mounted on a lattice and every periodic lattice structure belongs to one of the 230 possible space groups [3]. Some of these structures provide a low-energy effective theory for the electrons that is mathematically equivalent to the Dirac theory (e.g. in graphene [4]) or deformations of the Dirac theory as in certain structures of borophene [5,6] or the organic compound $\alpha-(\text{BEDT-TTF})_2\text{I}_3$ [7–12] or certain deformations of graphene [13]. For the Dirac electrons in solids, the dispersion relation giving energy $\varepsilon$ of the quasiparticles at every momentum $\vec{k}$ will be a *Dirac* cone, and enjoys an *emergent Lorentz invariance* at low energies or equivalently at length scales much larger than the lattice spacing [4].

In materials hosting a tilted Dirac cone, the cone-shaped dispersion relation is tilted in energy-momentum space [13,14]. This tilt is characterized by a set of anisotropy parameters $\zeta$.[1] It turns out that this particular form of anisotropy is very peculiar in the sense that tilting the Dirac cones is equivalent to attaching *vielbein* to Dirac fermions, and therefore the tilt parameter $\zeta$ can be nicely encoded into an emergent spacetime structure [15–19]. Therefore, in the same way that Dirac materials command an emergent Lorentz symmetry arising from an emergent Minkowski spacetime at long wavelengths, those materials that host a tilted Dirac cone can be assigned an emergent spacetime structure at long length scales[2] which is given by the metric

$$ds^2 = -v_F^2 dt^2 + (d\vec{r} - \zeta v_F dt)^2, \tag{1}$$

where $v_F$ is the velocity scale that defines the emergent spacetime. This upper limits of fermion velocity plays a role similar to the speed $c$ of light in high energy physics. However, note that $v_F$ in real materials is two or three orders of magnitude smaller than $c$. In the above relation, $\zeta$ is the tilt of the Dirac cone [19]. As can be seen, the role assumed by $\zeta$ in the above equation is

---

[1]Note that we denote vectors with $\vec{r}$ etc, while the tilt parameters are denoted by $\zeta$ rather than $\vec{\zeta}$ to emphasize that they are merely parameters of the spacetime, and not necessarily vectors in the new spacetime.

[2]Of course "long length scales" means, large compared to lattice scale.

significantly richer than a simple anisotropy of the space alone. In fact, this metric is associated with the long-distance structure of the underlying (in the case of the 8pmmn borophene) lattice and does not depend on whether the particles (excitations) are being studied are fermionic or bosonic [20, 21].

In this paper, we use the hydrodynamic theory as a generic low-energy effective description to bring up gross novel solid-state transport phenomena that can arise from an emergent metric (1) that is expected to be independent of many microscopic details. The emergence of hydrodynamic behavior in solids requires strong electron-electron interaction. As long as Fermi-liquid-like fixed points are concerned and effective quasiparticles continue to exist, such an emergent spacetime structure is expected to persist. This is simply because the dressed quasiparticles still roam about on the same underlying lattice structure. Therefore, the necessary conditions for the relevance of our hydrodynamic theory in the background metric (1) are: (i) the underlying lattice structure is not changed, (ii) there exists well defined fermionic quasiparticles. One might wonder, how restrictive are the above conditions. The condition (i) is valid as long as there is not structural phase transition or the lattice is not melted. To discuss condition (ii), we refer to the example of graphene. The direct observation of the Dirac cones in graphene – where the Coulomb interactions are strong enough to lead to hydrodynamic flow – is understood in terms of the renormalization of the Coulomb interactions. The key element is the lack of back-scattering for Dirac fermions that generates a "free Dirac fixed point" where the structure of spacetime is still Minkowski. Similar conditions hold for 2+1 dimensional tilted Dirac cone systems where the tilt parameter may be renormalized. Given the anisotropy arising from the tilt $\zeta$, the renormalization may be different in different directions. The validity of Minkowski spacetime in strongly correlated regime of graphene [22] leads us to assume that the structure of the spacetime given by (1) is likely to be preserved even in the hydrodynamic regime where the interactions produce the dominant scattering rates. For the rest of the paper, we will set $v_F = 1$ and will restore it when needed.

The isometries of the spacetime (1) are appropriate deformations of the Lorentz group [15]. As such, the spacetime defined by Eq. (1) is a deformation of the Minkowski spacetime by a continuous parameter $\zeta$. The presence of tilt $\zeta$ modifies many of the physical properties of the materials, in particular including their interfaces with superconductors [23, 24]. The above metric possesses a black hole horizon [25] that stems from spatial variation of the Galilean boost $\zeta$ in Eq. (1). Spatial variation of a parameter similar to $\zeta$ can emulate a black-hole horizon in atomic Bose-Einstein condensates [26], as well as in the polariton superfluids [27, 28]. The spin-polarized currents are also predicted to be associated with black-hole horizon for magnons [29]. Our proposal for a spacetime structure based on tilted Dirac fermions differs from the above systems and even from the tilted Dirac cone of Majorana fermions [30] in that (i) the tilted Dirac/Weyl systems required for our purpose are at ambient conditions and normal (non-superconducting) state and (ii) the quasiparticles of the theory are fermions, namely electrons and holes which carry electric charges. As such, any effects arising from the emergent structure of the spacetime, will leave direct signature in almost any electron spectroscopy experiment, including of course the transport (conductivity) phenomena discussed in this paper.

The hydrodynamic theory employed in this paper as the technical tool to calculate the transport properties of the electron fluid [31] is a generic theory that has been specialized to the tilted Dirac cone materials where the spacetime is given by metric (1). Hydrodynamic is an effective long-time and long-distance description of quantum many body systems that focuses on few conserved collective variables rather than embarking on the formidable task of addressing all the microscopic degrees of freedom. This powerful theory has many applications in various systems differing in microscopic details which are, however, described by the same equations. Only gross symmetry properties have to be properly incorporated into

the formulation of hydrodynamic for a system at hand. Applications of hydrodynamics approach in high energy physics includes quark-gluon plasma [32, 33], parity violation [34], chiral anomaly [35] and dissipative superfluid [36]. Within the hydrodynamic approach, one can come up with universal and model-independent predication such as kinematic viscosity value [37, 38]. The hydrodynamics approach can also be applied to fluid-gravity correspondence [39] which relate the dynamics of the gravity side to the hydrodynamic equations where the hydrodynamic fluctuation mode describes the fluctuations of black holes [40, 41].

In this work, we will be interested in a much simpler version of hydrodynamics in a space-time structure in $2 + 1$ dimensions subject to the metric (1) that describes electrons in sub-eV energy scales in solids with tilted Dirac cone. Let us announce our result in advance: In any conductor, the spatial gradient of temperature can generate heat and electric currents. The mingling of the space *and* time in Eq. (1) will allow the materials with tilted Dirac cone to generate heat and electric currents from *pure temporal* gradients.

The roadmap of the paper is as follows: In section 2 following Lucas [22] we formulate the hydrodynamics for the metric (1) as a natural, but categorically different generalization of the hydrodynamic theory of graphene. In section 3, in addition to the usual *tensor* transport coefficients of standards solids, we introduce *vector* transport coefficients required for a consistent treatment of the hydrodynamics in spacetime (1) where we discuss perfect fluid. It is followed by a treatment of the viscous fluids in section 4 in this new spacetime. We end the paper with a discussion and summary of the paper in section 5.

## 2 Hydrodynamics of tilted Dirac fermions

In this section we develop the hydrodynamics of electrons in a tilted Dirac cone material. For a planar material in $d = 2$ space dimensions, there is a two-parameter family of tilt deformations given by $\zeta = (\zeta_x, \zeta_y)$ to the Dirac equation in three dimensional spacetime. These deformations and the corresponding dispersion relation can be obtained if instead of the conventional Lorentz metric $\eta_{\mu\nu}$ one applies the following metric tensor

$$g_{\mu\nu} = \begin{pmatrix} -1 + \zeta^2 & -\zeta_j \\ -\zeta_i & \delta_{ij} \end{pmatrix}, \tag{2}$$

where $\zeta^2 = \zeta_x^2 + \zeta_y^2$ and $\delta$ is the $2 \times 2$ unit matrix [15, 19]. In this parameterization of the tilt we adopt the normalization $|\zeta| < 1$ so that spacetime makes sense in admissible coordinates. For convenience in the following computations, we introduce $\gamma = (1 - \zeta^2)^{-1/2}$. The Greek indices run over 0,1 and 2 for spacetime coordinates and the Latin indices run over spatial coordinates.

We assume that the electron-electron scattering rate $\tau_{ee}^{-1}$ is dominant over any other scattering rate such as electron-phonon ($\tau_{e-ph}^{-1}$) or electron-impurity scattering rate ($\tau_{e-imp}^{-1}$). This regime is attainable in graphene that hosts upright Dirac cone [42–45]. This has become possible by ability to tune the strength of electron-electron interaction via gating that sets the scale of the Fermi surface. In such a regime, an effective description at large distances ($\gg v_F \tau_{ee}$) and long time ($\gg \tau_{ee}$) is provided by the hydrodynamic equations given by conservation laws of Noether currents. Assuming translational invariance and gauge invariance of the underlying microscopic theory, there are conserved energy-momentum tensor $T^{\mu\nu}$ and an Abelian current vector $J^{\mu}$. They constitute nine independent components subject to four constraints $\partial_\mu T^{\mu\nu} = 0$ and $\partial_\mu J^\mu = 0$. We further assume that the spacetime has no torsion. The latter is valid for the emergent spacetime of tilted Dirac cone materials. Because in this case creation of torsion requires extra efforts.

In order to find unique solutions to hydrodynamics equation, it is assumed that the currents are determined through four auxiliary local thermodynamical quantities: the temperature $T(x)$, the chemical potential $\mu(x)$, a normalized time-like velocity vector field $u^\mu(x)$ (*i.e.* $u^\mu u_\mu = -1$) and their derivatives. The *fluid observer* moves along with the fluid and measures variables (local temperature, local mass density etc.) without ambiguities. The 3-velocity $u^\mu$ is defined relative to the Eulerian (arbitrary) observer. The fluid velocity $v^i$ is defined through $v^\mu = u^\mu/u^0$. The generalized Lorentz factor is defined as $\Gamma \equiv -n_\mu u^\mu = u^0$ where $n_\mu = (-1, \vec{0})$ is the time-like normal vector to the 2-space. An Eulerian observer attributes this factor to matter moving in the fluid frame. For instance, given the temperature measured by the fluid observer $T$, an Eulerian observer finds $T_E = \Gamma T$.

With respect to an arbitrary vector, any tensor can be decomposed to its transverse and longitudinal components. We have a freedom to identify $u^\mu$ with the velocity of energy flow in the so-called Landau frame $u^\mu \sim T^{\mu\nu} u_\nu$. Moreover, $T$ and $\mu$ can be defined so that

$$u_\mu J^\mu = -n \quad \text{and} \quad u_\mu T^{\mu\nu} = -\epsilon u^\nu, \tag{3}$$

where $n$ is the number density of charge carriers and $\epsilon$ is the energy density. In this frame, the particle current and the energy-momentum tensor can be decomposed as follows [46, 47]

$$J^\mu = nu^\mu + j^\mu, \tag{4}$$
$$T^{\mu\nu} = \epsilon u^\mu u^\nu + P\mathcal{P}^{\mu\nu} + t^{\mu\nu}, \tag{5}$$

where $P$ is a scalar (related to pressure $p$, see the following), $j^\mu$, $\mathcal{P}^{\mu\nu}$ and $t^{\mu\nu}$ are transverse vector and tensors that satisfy $u_\mu j^\mu = u_\mu \mathcal{P}^{\mu\nu} = u_\mu t^{\mu\nu} = 0$. The tensor $\mathcal{P}$ is defined as

$$\mathcal{P}^{\mu\nu} = g^{\mu\nu} + u^\mu u^\nu, \tag{6}$$

which is called the projection tensor; it is symmetric and in general has a non-vanishing trace. The inverse metric is

$$g^{\mu\nu} = \begin{pmatrix} -1 & -\zeta_i \\ -\zeta_j & \delta_{ij} - \zeta_i \zeta_j \end{pmatrix}. \tag{7}$$

The remaining elements $P, j^\mu$ and $t^{\mu\nu}$ are determined in terms of the derivatives of the hydrodynamic variables and yield the constituent equations in the desired order in derivatives. At first order (in derivatives) hydrodynamics, we find

$$\begin{aligned} P &= p - \xi_B \partial_\mu u^\mu, \\ j^\mu &= -\sigma_Q T \mathcal{P}^{\mu\nu} \partial_\nu(\mu/T) + \sigma_Q \mathcal{P}^{\mu\nu} F_{\nu\rho} u^\rho, \\ t^{\mu\nu} &= -\eta \mathcal{P}^{\mu\rho} \mathcal{P}^{\nu\sigma} \left( \partial_\rho u_\sigma + \partial_\sigma u_\rho - g_{\rho\sigma} \partial_\alpha u^\alpha \right), \end{aligned} \tag{8}$$

where $p$ is pressure in the local rest frame, $\xi_B$ is the bulk viscosity, $\sigma_Q$ is the intrinsic conductivity and $\eta$ is the shear viscosity. In the above equations, $F_{\nu\rho}$ is a tensor in 2+1 dimensions induced by an external electromagnetic field in the bulk. In passing, we recall that the coefficients in zero-order hydrodynamics $\epsilon$, $p$ and $n$ are fixed by $T$, $\mu$ and the equation of state in equilibrium thermodynamics [48–50]. Moreover, the non-negative parameters $\xi_B$, $\sigma_Q$ and $\eta$ (the Wilsonian coefficients of the effective hydrodynamic theory) are either measured in experiments or determined from an underlying microscopic (quantum) theory.

## 3 Emergent vector transport coefficients in tilted Dirac system:

We define the response of the electric current $\vec{J}$ and the heat current $\vec{Q}$ to an external electric field $\vec{E}$, spatial gradient $\vec{\nabla}T$ and possibly temporal variation $\partial_0 T$ of temperature as follows

$$J^i(t) = \int dt' \left[ \sigma^{ij} E_j(t') - \alpha^{ij} \partial_j T(t') - \beta^i T(t') \partial_0 \frac{\mu(t')}{T(t')} \right], \tag{9}$$

$$Q^i(t) = \int dt' \left[ T \bar{\alpha}^{ij} E^j(t') - \bar{\kappa}^{ij} \partial^j T(t') - \mu \gamma^i T(t') \partial_0 \frac{\mu(t')}{T(t')} \right], \tag{10}$$

where $\sigma^{ij}$, $\alpha^{ij}$, $\bar{\alpha}^{ij}$ and $\bar{\kappa}^{ij}$ are the usual tensor response coefficients relating the electric and heat currents to spatial gradients of electrochemical potential or temperature [2]. Anticipating electric/heat currents in response to temporal gradient $\partial_0 T$, we have additionally introduced the *vector transport coefficients* $\beta^i$ and $\gamma^i$. All the above coefficients are functions of $t - t'$ as the external influence (such as temperature or electrochemical potential) are applied in the laboratory frame where the spacetime structure has no $\zeta$ and hence are subject to time-translational invariance.

We compute the above transport coefficients within hydrodynamics theory. We imagine that fluid is perturbed around its equilibrium state $\big($specified by $\mu_0$, $T_0$ and $u_0^\mu = (1 - \zeta^2)^{-1/2}(1, \vec{0})$ in the rest frame of the fluid exposed to $\vec{E}_0 = \vec{0}\big)$ by a slight amount parameterized as follows

$$\delta T(t, \vec{x}), \quad \delta u^\mu(t, \vec{x}) = \gamma\big(-\gamma^2 \vec{\zeta} \cdot \delta \vec{v}, \delta \vec{v}\big), \quad \delta \vec{E}(t, \vec{x}). \tag{11}$$

The linear response of the electric current is given by

$$\delta J^i = n\gamma \delta v^i - \sigma_Q \zeta^i \frac{\mu_0}{T_0} \partial_0 \delta T - \sigma_Q g^{ij}\left(\gamma \delta E_j - \frac{\mu_0}{T_0} \partial_j \delta T\right). \tag{12}$$

Moreover, the leading order perturbation in derivatives is

$$\begin{aligned}
\delta T^{0i} = \; & \gamma^2(\epsilon_0 + p_0)\delta v^i - \delta p \zeta^i \\
& + \eta \gamma^2 \zeta^2 \zeta^i (2\partial_0 \delta u_0 - g_{00} \partial_\alpha \delta u^\alpha) \\
& - \eta(\zeta^j \zeta^i + \gamma^2 \zeta^2 g^{ij})(\partial_0 \delta u_j + \partial_j \delta u_0 - g_{0j} \partial_\alpha \delta u^\alpha) \\
& + \eta \zeta^j g^{ik}(\partial_j \delta u_k + \partial_k \delta u_j - g_{jk} \partial_\alpha \delta u^\alpha) + \xi_B \zeta^i \partial_\alpha \delta u^\alpha.
\end{aligned} \tag{13}$$

Here $\delta p$ and $\delta v^i$ are changes in pressure and velocity caused by the probe (external) electric field or temperature gradients and are first order in the probe fields. Using the above equation, we can compute thermal conductivity through

$$\delta Q^i = \frac{1}{\gamma^2} \delta T^{0i} - \delta(\mu J^i). \tag{14}$$

An interesting feature of Eq. (8) in the tilted Dirac/Weyl materials is a genuine effect where temporal variations generate currents: We note that in a linear hydrodynamic theory the terms in the brackets in these equations are already first order, and therefore at this order $\mathcal{P}^{\mu\nu} \to g^{\mu\nu}$. Therefore the spatial component $J^i$ of the current (in addition to the first term $nu^i$) will acquire a contribution proportional to $\partial_0 \mu$ and $\partial_0 T$ which is accompanied by the factor $g^{i0}$ that is nothing but the tilt parameter $= -\zeta^i$. This effect is solely dependent on the tilt parameters $\zeta^i$ and vanishes for upright Dirac/Weyl systems where $\zeta^i = 0$. That is why, we extend the conventional thermoelectric coefficients in Eqns. (9) and (10) to account for this important observation by introducing additional transport coefficients $\beta^i$ and $\gamma^i$.

We solve the hydrodynamic equations to evaluate the electric current and thermal current in the presence of background electric field and temperature spatiotemporal gradients. At this leading order, the charge conservation $\partial_\mu J^\mu = 0$ implies

$$\partial_t\Big[\gamma\delta n - n\gamma^3\zeta_i\delta v^i + \frac{\mu_0}{T_0}\sigma_Q(\gamma^2\zeta^2\partial_0 - \zeta^i\partial_i)\delta T + \sigma_Q\zeta^i\gamma\delta E_i\Big] + \partial_i(\delta J^i) = 0. \quad (15)$$

The energy and momentum conservations are of the form of $\partial_\mu T^{\mu\nu} = 0$. In the presence of external influence the right hand side of the above equation will be non-zero to account for the transfer of energy-momentum to the system by external agents the handling of which requires some care and will be discussed shortly. For warmup, let us first consider a situation without external sources. In this case, the energy conservation $\partial_\mu T^{\mu 0} = 0$ reads

$$\partial_t\Big[\gamma^2\delta\epsilon + \gamma^2\zeta^2\delta p - 2\gamma^4(\epsilon_0 + p_0)\zeta_i\delta v^i - \eta\gamma^4\zeta^4(2\partial_0\delta u_0 - g_{00}\partial_\alpha\delta u^\alpha)$$
$$+2\eta\zeta^2\gamma^2\zeta^i(\partial_i\delta u_0 + \partial_0\delta u_i - g_{i0}\partial_\alpha\delta u^\alpha) - \eta\zeta^i\zeta^j(\partial_i\delta u_j + \partial_j\delta u_i - g_{ij}\partial_\alpha\delta u^\alpha) - \xi_B\gamma^2\zeta^2\partial_\alpha u^\alpha\Big]$$
$$+\partial_i\delta T^{i0} = 0, \quad (16)$$

whereas the momentum conservation $\partial_\mu T^{\mu i} = 0$ is[3]

$$-\frac{\epsilon_0 + p_0}{\tau_{\text{imp}}}\delta u^i = \partial_0(\delta T^{0i}) + \partial_i\Big[\delta p g^{ij} - \eta\zeta^i\zeta^j(2\partial_0\delta u_0 - g_{00}\partial_\alpha\delta u^\alpha)$$
$$+\eta(\zeta^i g^{jk} + \zeta^j g^{ik})(\partial_0\delta u_k + \partial_k\delta u_0 - g_{0k}\partial_\alpha\delta v^\alpha)$$
$$-\eta g^{il} g^{kj}(\partial_l\delta u_k + \partial_k\delta u_l - g_{lk}\partial_\alpha u^\alpha) - \xi_B g^{ij}\partial_\alpha\delta v^\alpha\Big], \quad (17)$$

where the parameter $\tau_{\text{imp}}$ is the relaxation time due to scattering from impurities. The above expressions in the square bracket are leading order perturbations in number density, energy, density and the stress tensor. We find the response of equations (15), (16) and (17) to external perturbations in order to extract transport coefficients.

## 3.1 External forces in the tilt geometry

In this section, we include external forces which modifies the right hand side of the conservation equations for the energy-momentum tensor. First, we consider the electromagnetic fields and spatial gradients of the temperature. Then we discuss how to include temporal gradients.

### 3.1.1 Electromagnetic force and spatial gradients of temperature

The electromagnetic force can be introduced through the following coupling to the electron current

$$\partial_\nu T^{\mu\nu} = F^{\mu\nu}J_\nu. \quad (18)$$

Although Eq. (18) looks covariant, we note that the electromagnetic field (the photon) propagates in Minkowski space-time and is not affected by the tilt parameters, whereas the electron current confided to the sample is affected by the tilt parameter.

The Euclidean time has period $\frac{1}{T}$ and we rescale the time coordinate as $t = \frac{t'}{T}$ [51, 52]. Then, the metric elements are $g_{t't'} = -\frac{1}{\gamma^2 T^2}$ and $g_{t'i} = -\frac{\zeta^i}{T}$. A small temperature gradient,

---

[3]We note that, due to electron scattering off impurities and phonons, momentum is not a conserved charge for electron fluid in metals. Thus, we need to introduce a term which is responsible for momentum relaxation. In low temperature, we only consider the dominant electron scattering off impurities.

$T \to T + x^i \partial_i T$, implies that

$$\delta g_{t't'} = \frac{2x^i \partial_i T}{\gamma^2 T^3}, \tag{19}$$

$$\delta g_{t'i} = \frac{\zeta^i x^j \partial_j T}{T^2}. \tag{20}$$

Taking into account gauge transformations on the background field $\delta g_{\mu\nu} = \nabla_\mu \alpha_\nu + \nabla_\nu \alpha_\mu$, $\delta A_\mu = A_\nu \partial_\mu \alpha^\nu + \alpha^\nu \partial_\nu A_\mu$ and assuming the time dependence as $e^{-i\omega' t'}$, we take the the diffeomorphism parameter $\alpha_\mu$ as $\alpha_{t'} = \frac{-i x^i \partial_i T}{\omega' T^3} e^{-i\omega' t'}$ and $\alpha_i = \frac{-i x^j \partial_j T \zeta_i}{T^2 \omega'} e^{-i\omega' t'}$ that gives

$$\delta g_{tt} = 0, \tag{21}$$

$$i\omega \delta g_{ti} = \frac{\partial_i T}{\gamma^2 T}, \tag{22}$$

$$i\omega \delta g_{ij} = \frac{\zeta_i \partial_j T + \zeta_j \partial_i T}{T}, \tag{23}$$

$$i\omega \delta A_i = -\mu \frac{\partial_i T}{T}, \tag{24}$$

where quantities are scaled back to the original time coordinate $t$. Under the above gauge transformation, the effective 2+1 dimensional action changes as

$$\begin{aligned}
\delta S &= \int d^2 x \, dt \sqrt{-g} (T^{\mu\nu} \delta g_{\mu\nu} + J^\mu \delta A_\mu) \\
&= \int d^2 x \, dt \sqrt{-g} (\frac{1}{\gamma^2} T^{0i} - \mu J^i) \frac{-\partial_i T}{-i\omega T} + T^{ij} \delta g_{ij} + J^i \frac{E_i}{i\omega}),
\end{aligned} \tag{25}$$

which in particular suggests that the thermal current is $Q^i = \frac{1}{\gamma^2} T^{0i} - \mu J^i$. Moreover, it can be seen that in the tilted space-time 2, we have a momentum flow which is coupled to $\delta g_{ij}$ which is determined by the temperature gradient.

In colculsion, we promote partial derivative to covariant derivative and write the hydrodynamic Eq. (18) in a non-trivial background metric (2) perturbed by perturbations (21) to (24) and keep the leading order terms. Doing so the energy and momentum equations for the perfect fluid become

$$\partial_\mu T^{\mu 0} = n_0 \gamma \zeta_i E^i - [(\epsilon_0 + p_0) - \mu n_0 \gamma] \frac{\zeta^i \partial_i T}{T}, \tag{26}$$

$$\partial_\mu T^{\mu i} = \left[ (\epsilon_0 + p_0) - \frac{\mu n_0}{\gamma} \right] \frac{\partial_i T}{T} - (\epsilon_0 + p_0) \zeta^i \frac{\zeta^j \partial_j T}{T} + \frac{n_0 E^i}{\gamma}. \tag{27}$$

### 3.1.2 Temporal gradient of temperature

Following the same line of the reasoning as above, the response of the system to the temporal gradient of temperature $T \to T + t' \partial_{t'} T$ in the re-scaled time coordinate implies

$$\delta g_{t't'} = \frac{2t' \partial_{t'} T}{\gamma^2 T^3}, \tag{28}$$

$$\delta g_{t'i} = \zeta^i \frac{t' \partial_{t'} T}{T^2}. \tag{29}$$

Then, in the oroginal time coordinate $t$ we find

$$\partial_t \delta g_{tt} = \frac{2\partial_t T}{\gamma^2 T}, \tag{30}$$

$$\partial_t \delta g_{ti} = \zeta^i \frac{\partial_t T}{T}. \tag{31}$$

Consequently, the presence of temporal gradient of temperature modifies the right hand side of the energy and momentum conservation equations according to

$$\partial_t T^{tt} = 2\gamma^2(\epsilon_0 + p_0)\frac{\partial_t T}{T}, \tag{32}$$

$$\partial_t T^{0i} = -\frac{\epsilon_0 + p_0}{\tau_{\text{imp}}}\gamma \delta v^i. \tag{33}$$

## 3.2  Perfect fluid: Non-Drude features

To study the perfect fluid, we start by ignoring dissipation $\eta = \xi_B = 0$, Furthermore, we are interested in a homogeneous flow; *i.e.* spatially uniform solutions with $\partial_j \delta v^i = 0$ and $\partial_i \delta p = 0$. With these assumptions, the energy-momentum equations become

$$\partial_t\Big[\gamma^2(\delta\epsilon + \zeta^2\delta p) - 2\gamma^4(\epsilon_0 + p_0)\zeta_j\delta v^j\Big] = n_0\gamma\zeta_i E^i - [(\epsilon_0 + p_0) - \mu n_0\gamma]\frac{\zeta^i \nabla_i T}{T}, \tag{34}$$

$$\partial_t\Big[\gamma^2(\epsilon_0 + p_0)\delta v^i - \zeta^i\delta p\Big] = -\frac{\epsilon_0 + p_0}{\tau_{\text{imp}}}\gamma \delta v^i$$
$$+ \Big[(\epsilon_0 + p_0) - \frac{\mu n_0}{\gamma}\Big]\frac{\nabla_i T}{T} - (\epsilon_0 + p_0)\zeta^i\frac{\zeta^j \partial_j T}{T} + \frac{n_0 E^i}{\gamma}. \tag{35}$$

After a Fourier transformation in time, the above equations can be solved for the velocity perturbation as

$$\delta v^i = (\mathcal{C}^{-1})^i_j\Big[n_0 A^{jk}E^k + B^{jk}\nabla_k T\Big], \tag{36}$$

where $A^i_j$ and $B^i_j$ are defined by

$$A^{ij} = \frac{1}{\gamma}\Bigg[\delta^{ij} - \frac{\zeta^i \zeta^j}{2 + \zeta^2}\Bigg], \tag{37}$$

$$B^{ij} = \frac{1}{T}\Bigg[\Big(\epsilon_0 + p_0 - \frac{\mu n_0}{\gamma}\Big)\delta^{ij} - \Big((\epsilon_0 + p_0)\frac{3 + \zeta^2}{2 + \zeta^2} - \frac{\mu n_0\gamma}{2 + \zeta^2}\Big)\zeta^i \zeta^j\Bigg], \tag{38}$$

and $\mathcal{C}^{-1}$ is the inverse matrix of

$$\mathcal{C}^i_j = \Bigg[\gamma\frac{\epsilon_0 + p_0}{\tau_{\text{imp}}}(1 - i\gamma\omega\tau_{\text{imp}})\Bigg]\delta^i_j + \Bigg[2i\omega\gamma^2\frac{\epsilon_0 + p_0}{2 + \zeta^2}\Bigg]\zeta^i \zeta_j, \tag{39}$$

which is explicitly computed in the appendix. Then, we compute the electric current (12) as a response to spatiotemporal variations of the electric field and temperature. Finally, by applying equation (36) we can read the coefficients in (9) as follows

$$\sigma^{ij} = \gamma n_0^2(\mathcal{C}^{-1})^i_k A^{kj} + g^{ij}\gamma\sigma_Q, \tag{40}$$

$$\alpha^{ij} = -\gamma n_0(\mathcal{C}^{-1})^i_k B^{kj} - \frac{\mu_0}{T_0}\sigma_Q g^{ij}. \tag{41}$$

The pole structure of the electric and heat conductivity tensors are the same. This is because they are both related to the determinant of the matrix $\mathcal{C}$. Therefore, we focus on the poles of the conductivity. We find that the conductivity tensor has the following two poles

$$\omega_1 = \frac{-i}{\gamma \tau_{\text{imp}}} \equiv \frac{-i}{\tau_{\text{imp}}^{\text{lab}}}, \tag{42}$$

$$\omega_2 = \frac{2 + \zeta^2}{2 - \zeta^2} \omega_1. \tag{43}$$

In the upright Dirac cone with Minkowski spacetime structure, $\zeta \to 0$, both poles $\omega_1$ and $\omega_2$ of Eqns. (42) and (43) reduce to the Drude result. To understand these poles, we have defined a *redshifted relaxation time* $\tau_{\text{imp}}^{\text{lab}} = \gamma \tau_{\text{imp}}$. In this definition $\tau_{\text{imp}}$ can be interpreted as the microscopic relaxation time experienced by electrons in the spacetime with a given $\zeta$, while $\tau_{\text{imp}}^{\text{lab}}$ can be interpreted as the same time measured in the laboratory by the experimentalist sitting in the laboratory spacetime having $\zeta = 0$. As such, the pole at $\omega_1$ is a natural extension of the Drude pole to the geometry (1). However, the new pole $\omega_2$ arises from the new spacetime structure. Although at $\zeta \to 0$ limit it becomes degenerate with the Drude pole, but at $\zeta \to 1$ it can become up to 3 times larger than $\omega_1$. Both poles are on the imaginary axis, and their real part is zero, as they are caused by very low-energy (Drude) excitations across the Fermi level. Then the question will be, is there a way to distinguish the contributions from the Drude pole $\omega_1$ and the spacetime pole $\omega_2$? To answer this question, we need to look at the residues at the two poles that determine the spectral intensity associated with each pole.

Let us start by looking at the residue of the absorptive part of the conductivity, namely the longitudinal conductivity $\sigma^{xx}$. The residues at the above poles can be written in the following suggestive form[4]

$$\text{Res}(\sigma_{\omega_1}^{xx}) = \frac{n^2 v_F^2}{\epsilon_0 + p_0} \frac{\zeta_y^2}{\gamma^2 \zeta^2}, \tag{44}$$

$$\text{Res}(\sigma_{\omega_2}^{xx}) = \frac{n^2 v_F^2}{\epsilon_0 + p_0} \frac{2\zeta_x^2}{\gamma^2 (2 - \zeta^2) \zeta^2}. \tag{45}$$

The above form suggests that the sum of the intensities of the two poles has a very well defined $\zeta \to 0$ limit given by $n^2 v_F^2 / (\epsilon_0 + p_0)$. This observation combined with the fact that the $\zeta \to 0$ limits of $\omega_2$ gives $\omega_1$ implies that in the presence of the tilt, the Drude peak splits into two peaks. The new pole $\omega_2$ that we call it "spacetime pole" is an offspring of Drude pole.[5] By symmetries of space time $\sigma^{yy}$ can be extracted from $\sigma^{xx}$ only by exchanging $\zeta_x \leftrightarrow \zeta_y$. So there is no new information in residues of the poles of $\sigma^{yy}$.

As pointed out, both $\omega_1$ and $\omega_2$ poles are on the imaginary axis and their real part is zero. To disentangle their contribution, note that the meaning of the longitudinal conductivity $\sigma^{xx}$ is the current along the applied electric field (both assumed along the $x$ direction). The $x$ axis can subtend an angle $\theta$ with the tilt direction $\zeta$. This can be an interesting variable. Therefore, in Fig. 1 we have plotted the dependence of the residues of the longitudinal part of the conductivity on the polar angle $\theta$ of the tilt direction. The solid (dashed) lines correspond to the Drude-like pole $\omega_1$ (spacetime pole $\omega_2$). Various colors correspond to different tilt magnitudes. Both poles have a bipolar pattern. But their nodal structure is different which helps to identify which pole is contributing the spectral weight. When the electric field is

---

[4]In all plots of this paper, the Fermi velocity $v_F$ has been explicitly included and expressed in units of $[r/\tau_{\text{imp}}]$ where $r$ is a length scale that defines the average linear dimension available for one electron and is related to the density by $nr^2 = 1$. For typical $n \sim 10^{12} \text{cm}^{-2}$, we get $r = 10^{-6}$cm. Furthermore a typical value of $\tau_{\text{imp}}$ is $10^{-13}s$.

[5]However, one has to note that in the limit $\zeta \to 0$, a zero at $\omega_1$ appears in the numerator of conductivity coefficients that prevents the formation of second order Drude pole.

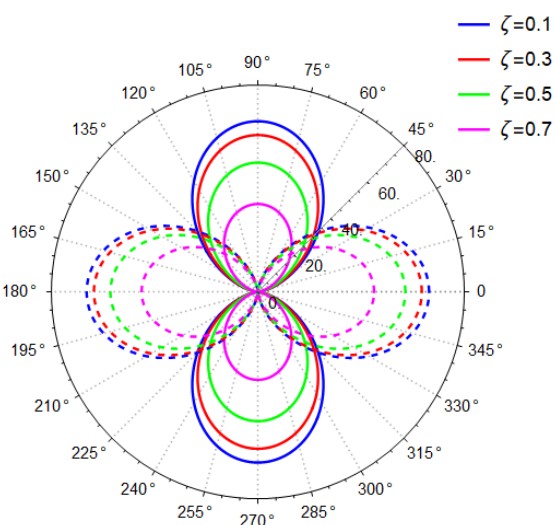

Figure 1: Polar dependence of the residue of the longitudinal conductivity $\sigma^{xx}$ for Drude like pole $\omega_1$ (solid lines) and spacetime pole $\omega_2$ (dashed lines) in units of $[\text{meV}.\tau_{\text{imp}}^2]^{-1}$ . *The structure of spacetime is such that the new pole $\omega_2$ is born from a parent Drude pole $\omega_1$ in the sense that the sum of their intensities has a well defined $\zeta \to 0$ limits.* The polar angle is measured from the direction of the tilt vector $\zeta$.

applied along the tilt direction ($\theta = 0$), the residue of the Drude-like pole vanishes and the absorption is entirely contributed by the offspring pole $\omega_2$. By rotating the applied electric field away from the tilt direction, the Drude-like pole $\omega_1$ takes over the spacetime pole $\omega_2$. When the applied electric field is completely perpendicular to $\zeta$, the entire absorption is contributed by the $\omega_1$ (Drude) pole. The common feature of solid and dashed curves in Fig. 1 is that the spectral weight of both the parent Drune pole $\omega_1$ and its offspring pole $\omega_2$ decreases by increasing the tilt $\zeta$.

## 3.3 Tilt-induced Hall-like response

Now we would like to see how does the spacetime structure affect the off-diagonal (Hall-like) conductivity. In a Hall bar setup experiment, a background magnetic field along the $z$-direction is assumed. Then a current driven along $x$ axis gives rise to a voltage drop along $y$ axis. This effect is quantified by the off-diagonal component $\sigma^{xy}$ (or $\sigma_H$). Similarly a current driven along $y$ direction will give a voltage drop in $-x$ direction. As such, in the standard Hall setup, the conductivity tensor is fully antisymmetric. Therefore *the rotational invariance implies that* $\sigma^{xy} = -\sigma^{yx}$. If the conductivity tensor had any symmetric part, there exists a coordinate system in which the symmetric part of the off-diagonal conductivity becomes zero. However, the assumption of rotational invariance implies that, all the coordinate systems related by a rotation must be equivalent. Therefore, to be consistent, the rotational invariance prohibits symmetric parts for the off-diagonal conductivity. Now let us consider the case of tilted Dirac cone materials. As pointed out in the introduction, the presence of the tilt $\zeta$ breaks that rotational invariance.[6]

The new spacetime structure indeed gives rise to a totally novel form of the *symmetric* part in the conductivity tensor. Unlike the ordinary spactime, such a symmetric part does not vanish, because the rotational invariance does not hold anymore. Furthermore, such a Hall-like

---

[6]In fact rotation is not the isometry of the metric (1) anymore. Solving Killing equation gives the correct isometries of the spacetime (1) which are extensions of the ordinary rotation and ordinary Lorentz boosts [15].

coefficient arises *in the absence of external magnetic field* and is rooted in the off-diagonal metric elements of the metric $g^{ij} \propto \zeta^i \zeta^j$. This element of the metric directly leads to a non-zero Hall-like electric and Hall-like thermal coefficient proportional to $\zeta^i \zeta^j$ arising in Eqs. (40) and (41). This effect also solely depends on the presence of the tilt $\zeta^i$. Mathematically speaking, in the absence of $\zeta^i$ (isotropic space), the conductivity tensor $\sigma^{ij}$ (as in the case of graphene) will become proportional to (the isotropic tensor) $\delta^{ij}$. But in the present case, the anisotropy of the space will be reflected in a $\zeta^i \zeta^j$ dependence in all tensorial quantities, including the electric conductivity and heat conductivity tensors. Since $\zeta^i \zeta^j = \zeta^j \zeta^i$, the above contribution from the spacetime structure to the off-diagonal transport coefficients will be always *symmetric*. This is what we wish to emphasize by using "Hall-like" response instead of "Hall" response. As such, in the tilted Dirac cones systems, we obtain a novel form of anomalous (i.e. without the need to externally applied B field) Hall response which is symmetric and can be directly attributed to the structure of the spacetime.

To summarize, the off-diagonal (Hall) response may have both antisymmetric and symmetric parts. But in materials with Minkowski or Gallilean spacetime, the symmetric part of the Hall response is inert in ordinary materials. But tilted Dirac cone materials provide a unique opportunity for the appearance of a symmetric (anomalous) Hall response.

Now let us consider the off-diagonal (transverse) component of the conductivity, namely $\sigma^{xy}$. The pole structure of off-diagonal components of the conductivity is the same as the diagonal part, as in both cases the poles are contributed by the determinant of the same matrix $\mathcal{C}$ given in Eq. (39). However, the residues of the off-diagonal response differ in a very interesting way from the diagonal conductivity and are given by

$$\text{Res}(\sigma^{xy}_{\omega_1}) = -\frac{n^2 v_F^2}{\epsilon_0 + p_0} \frac{\zeta_x \zeta_y}{\gamma^2 \zeta^2}, \tag{46}$$

$$\text{Res}(\sigma^{xy}_{\omega_2}) = \frac{n^2 v_F^2}{\epsilon_0 + p_0} \frac{2\zeta_x \zeta_y}{\gamma^2 (2 - \zeta^2) \zeta^2}. \tag{47}$$

Again the sum of the above residues has a clear $\zeta \to 0$ (Minkowski) limit where it vanishes. Being off-diagonal conductivity, the above poles can not be associated with absorption (dissipation). But still it is interesting to note that how a total zero pole intensity in the $\zeta \to 0$ limits evolves into two poles of opposite intensity upon deviation of the spacetime structure from the Minkowski limit. Mathematically, $\sigma^{ij}$ (and also $\alpha^{ij}$) being tensors are naturally expected to have a term proportional to $g^{ij} \sim \zeta^i \zeta^j$.

The residues of the Hall conductivity arising from $\omega_1$ and $\omega_2$ poles are plotted in Fig. 2. As can be seen both poles display quadrangular pattern. Also their behavior with $\zeta$ is similar. Both intensities decrease upon decreasing the tilt magnitude $\zeta$.

The conclusion is that, the longitudinal conductivity suffices to disentangle the role of Drude-like pole $\omega_1$ and spacetime pole $\omega_2$ in the conductivity of tilted Dirac material sheets.

## 3.4 Heat current from gradual heating

Now let us turn our attention to the heat (energy) transport coefficients. The thermal current in a non-viscous homogeneous fluid is given by

$$Q^i = \frac{1}{\gamma^2} (\gamma^2 (\epsilon_0 + p_0) \delta v^i - \zeta^i \delta p) - \mu_0 \delta J^i, \tag{48}$$

where $\delta J^i$ is given in (12). We analyze the energy-momentum conservation equations for an externally applied constant spatio-temporal temperature gradient. For the spatial temperature

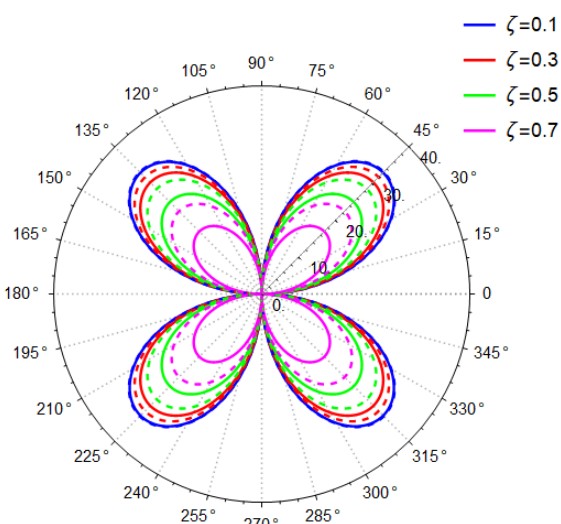

Figure 2: Polar angle dependence of the residues of $\sigma^{xy}$ for $\omega_1$ (solid lines) and $\omega_2$ (dashed lines) in units of $[\text{meV}.\tau_{\text{imp}}^2]^{-1}$. The polar angle denotes the angle between the applied electric field and the tilt vector $\zeta$. The sign of the two residues are opposite as in (46) and (47).

gradient, the energy-momentum conservation equations are

$$\partial_t\left[\gamma^2(\delta\epsilon + \zeta^2\delta p) - 2\gamma^4(\epsilon_0 + p_0)\zeta_j\delta v^j\right] = -[(\epsilon_0 + p_0) - \mu n_0\gamma]\frac{\zeta^i\partial_i T}{T},$$

$$\partial_t\left[\gamma^2(\epsilon_0 + p_0)\delta v^i - \zeta^i\delta p\right] = -\frac{\epsilon_0 + p_0}{\tau_{\text{imp}}}\gamma\delta v^i + \left[(\epsilon_0 + p_0) - \frac{\mu n_0}{\gamma}\right]\frac{\partial_i T}{T} - (\epsilon_0 + p_0)\zeta^i\frac{\zeta^j\partial_j T}{T}.$$

In $d = 2$ space dimensions, the above equations are $d + 1$ equations for a total of $d + 1$ unknowns. These are $d$ velocity perturbation $\delta v^j$ and one pressure perturbation $\delta p$. To be more intuitive, we solve these equations in the real space. Using the first of the above equations to express $\partial_t\delta p$ in terms of $\partial_t\delta v^i$ and $\partial_j T$ gives

$$\partial_t\delta p = \frac{2\gamma^2(\epsilon_0 + p_0)\zeta_j\partial_t\delta v^j}{2 + \zeta^2} - \frac{\epsilon_0 + p_0 - \mu n_0\gamma}{\gamma^2(2 + \zeta^2)}\frac{\zeta^j\partial_j T}{T}, \tag{49}$$

which can be used in the second equation above to eliminate $\delta p$ that results in the following equation for the velocity perturbations $\delta v^i$:

$$M^{ij}\partial_t\delta v^j + \frac{1}{\gamma\tau_{\text{imp}}}\delta v^i = N^{ij}\frac{\partial_j T}{T}, \tag{50}$$

where the $2 \times 2$ matrices $M^{ij}$ and $N^{ij}$ are defined by

$$M^{ij} = \delta^{ij} - \frac{2\zeta^i\zeta^j}{2 + \zeta^2}, \tag{51}$$

$$N^{ij} = \frac{(\epsilon_0 + p_0)(2 + \zeta^2)\delta^{ij} - (\gamma^2(\epsilon_0 + p_0)(2 + \zeta^2) + \epsilon_0 + p_0 - \mu n_0\gamma)\zeta^i\zeta^j}{\gamma^4(\epsilon_0 + p_0)(2 + \zeta^2)}. \tag{52}$$

The general solutions of Eq. (50) is

$$\delta v^i = \gamma\tau_{\text{imp}}N^{ij}\frac{\partial_j T}{T} + \left[c_1 e^{\frac{-\lambda_1 t}{\gamma\tau_{\text{imp}}}}L_1^i + c_2 e^{\frac{-\lambda_2 t}{\gamma\tau_{\text{imp}}}}L_2^i\right], \tag{53}$$

where $\lambda_a$ and $\vec{L}_a$ for $a = 1, 2$ are two eigenvalues and eigenvectors of the matrix $M^{-1}$ and $c_a$ are some constants determined by the initial conditions. We see that in the late time[7] $t \gg \tau_{\text{imp}}$, the transient terms in the square brackets fade away and velocity $\delta v^i$ becomes a constant given by the first term of the above equation. Substitution of the above result in Eq. (49) gives the following late time behavior for $\partial_t \delta p$

$$\partial_t \delta p = -\frac{\epsilon_0 + p_0 - \mu n_0 \gamma}{\gamma^2 (2 + \zeta^2)} \frac{\zeta^j \partial_j T}{T}, \tag{54}$$

so the late time behavior is $\delta p \propto t$. Plugging this result in Eq. (48) implies a similar $t$-linear behavior for the thermal current $Q^i$. This result simply states that if an external source manages to maintain a constant spatial temperature gradient, the resulting heat current $Q^i$ will be directed along $\zeta^i$ and is $t-$linear. This amounts to a constant thermal power. This behavior is expected in a generically anisotropic medium where the energy supplied by the external source that maintains the constant spatial gradient oriented along the preferred direction $\zeta$ of the material. It may sound a bit unusual for the spatial gradient of temperature to induce a time-dependent fluctuation of the pressure. The origin of this can be traced back to the assumption of zero spatial pressure gradient, $\partial_i \delta p = 0$. To see how this happens let us relax this assumption that will immediately result in a steady-state solution of energy-momentum equation as follows:

$$\partial_t \left[ \gamma^2 (\delta \epsilon + \zeta^2 \delta p) - 2\gamma^4 (\epsilon_0 + p_0) \zeta_j \delta v^j \right] + \partial_i \left[ \gamma^2 (\epsilon_0 + p_0) \delta v^i - \zeta^i \delta p \right]$$

$$= -[(\epsilon_0 + p_0) - \mu n_0 \gamma] \frac{\zeta^i \partial_i T}{T}, \tag{55}$$

$$\partial_t \left[ \gamma^2 (\epsilon_0 + p_0) \delta v^i - \zeta^i \delta p \right] + g^{ij} \partial_j \delta p = -\frac{\epsilon_0 + p_0}{\tau_{\text{imp}}} \gamma \delta v^i$$

$$+ \left[ (\epsilon_0 + p_0) - \frac{\mu n_0}{\gamma} \right] \frac{\partial_i T}{T} - (\epsilon_0 + p_0) \zeta^i \frac{\zeta^j \partial_j T}{T}. \tag{56}$$

To find the steady-state solution, one simply sets all the partial time derivatives to zero. We further assume that in the late time limit, velocity will be a constant. Then we obtain

$$\zeta^i \partial_i \delta p = \left( \epsilon_0 + p_0 - \mu n_0 \gamma \right) \frac{\zeta^i \partial_i T}{T}, \tag{57}$$

$$\frac{\epsilon_0 + p_0}{\tau_{\text{imp}}} \delta v^i = -g^{ij} \partial_j \delta p + \left[ \epsilon_0 + p_0 - \frac{\mu n_0}{\gamma} \right] \frac{\partial_i T}{T} - (\epsilon_0 + p_0) \zeta^i \frac{\zeta^j \partial_j T}{T}. \tag{58}$$

Therefore the assumption of $\partial_t \delta p = 0$ inevitably leads to a steady state soluiton with $\partial_i \delta p \neq 0$. However, mathematically, there exists a non-steady-state solution as well that relies on the assumption of $\partial_i \delta p = 0$. In this case, a non-zero $\partial_t \delta p$ develops that is given by Eq. (54).

The peculiar form of anisotropy that sets the off-diagonal time-space components of the metric (7) is capable to provide the above form of accumulative heat current in response to temporal gradient of the temperature as well. To realize this fascinating result, one needs to incorporate the temporal gradient of temperature, $\partial_t T$ as a source. Therefore the right hand side of energy and momentum conservation will be replaced by terms in Eq. (32) and (33) so that now the perturbations $\delta v^i$ and $\delta p$ satisfy linear equations

$$\partial_t \delta p = \frac{2\gamma^2 (\epsilon_0 + p_0)}{2 + \zeta^2} \zeta_j \partial_t \delta v^i + \frac{2(\epsilon_0 + p_0)}{2 + \zeta^2} \frac{\partial_t T}{T}, \tag{59}$$

$$M^{ij} \partial_t \delta v^j + \frac{1}{\gamma \tau_{\text{imp}}} \delta v^i = \frac{2}{(2 + \zeta^2) \gamma^2} \zeta^i \frac{\partial_t T}{T}. \tag{60}$$

---

[7]Eigenvalues $\lambda_1$ and $\lambda_2$ are positive, so the late time is well defined.

Similar to the case of sourcing with $\partial_j T$, the general solutions of the above equation are

$$\delta v^i = \frac{2\gamma\tau_{\text{imp}}}{(2+\zeta^2)\gamma^2}\zeta^i\frac{\partial_t T}{T} + \left[c_1 e^{\frac{-\lambda_1 t}{\gamma\tau_{\text{imp}}}}L_1^i + c_2 e^{\frac{-\lambda_2 t}{\gamma\tau_{\text{imp}}}}L_2^i\right]. \tag{61}$$

Again in the late time limit the transient terms in the bracket fade away and the first term of the above equation survives. Substitution in the pressure equation gives the following late time pressure perturbation

$$\partial_t \delta p = \frac{2(\epsilon_0 + p_0)}{2+\zeta^2}\frac{\partial_t T}{T}. \tag{62}$$

Once again, plugging in Eq. (48) gives a thermal current directed along $\zeta^i$. The essential difference of the heat current obtained by the pressure (62) with respect to (54) is that here the heat current is caused by *temporal variations of temperature*. This is a fundamentally new concept in the generation of electron current. In the so called hot deserts, it is not easy to control the spatial gradient of temperature as they are determined by complicated solutions of the surrounding air. But absorption of sun light from the coldest moment of the midnight to the hottest time of the mid-day can serve as a natural resource of $\partial_t T$. As such, the spacetime geometry in tilted Dirac cone materials qualifies them as a novel class of materials that can convert gradual heating to electron currents that carry heat. Since such a transport is only based on the spacetime parameter $\zeta$, it evenly couples to both electrons and holes. As such, the net electric current is zero, unless an asymmetry between electrons and holes is implemented.

In a finite sample, one must solve the hydrodynamics once again subject to appropriate boundary conditions. However, on physical grounds, one can speculate about such a situation as follows: Due to the heat current obtained above, the hot electrons accumulate on one side of the sample. As a result the accumulated density will generate an electric field that balances the force resulting from the linear increase of the pressure. This balance will equilibrate a finite system. But in open systems where the tilted Dirac material is part of a circuit, the accumulated electrons get released and supply heat current to the circuit. The heat current obtained in this way solely depends on the presence of a tilt vector $\zeta$, and it is present both for $\partial_j T$ and $\partial_t T$ sources. The latter effect has no analog in other solid-state systems. The peculiar $\zeta$ dependence along with a $t$-linear dependence of this particular form of heat current can be employed to separate it from the other terms in the heat current. The anomalous heat transport in 8$Pmmn$ borophene studied in Ref. [53] can be re-examined in the light of this new term.

## 3.5 Vector transport coefficients

So far we have seen that in materials with ordinary spacetime structure, the $\partial_t T$ has no chance to serve as a source of transport and as such, it is inert in most materials. But this source can be revived as we explained in the previous subsection. Now we can turn our attention to Eqs. (9) and (10) where new transport coefficients, $\beta^i$ and $\gamma^i$ are introduced that quantify the transport response of the system to temporal gradients of temperature and are given by

$$\beta^i = -\sigma_Q \zeta^i + \frac{2n_0 \tau_{\text{imp}} \zeta^i}{\mu(2+\zeta^2)}, \tag{63}$$

$$\gamma^i = \sigma_Q \zeta^i + \left[\frac{2(\epsilon_0+p_0)}{\mu^2\gamma^2(2+\zeta^2)}\left(1-\frac{t}{\gamma\tau_{\text{imp}}}\right) - \frac{2n_0\tau_{\text{imp}}}{\mu(2+\zeta^2)}\right]\zeta^i. \tag{64}$$

These *vector* transport coefficients (as opposed to tensor transport coefficients) relate the temporal gradients of electrochemical potential and temperature to electric and heat currents.

As can be seen from Eqns. (63) and (64), at the present order of calculations, these are proportional to the only available direction, namely $\zeta$. The first term in both (63) and (64) in addition to $\zeta$ are proportional to the ability $\sigma_Q$ of the electron system to conduct. This part of the vector coefficient is active when either of the electrons or holes dominate. When the chemical potential coincides with the Dirac node, the conduction ability $\sigma_Q$ of electrons and holes cancel each other. So to obtain a contribution from this term, one must ensure that the tilted Dirac material has its chemical potential away from the Dirac node. The second term in (63) and third term in Eq. 64 have similar origins and arise from $J^i$ of Eqs. (12) and (14). The second term in Eq. (64) originates from the $\delta T^{0i}$ of Eq. (14). The origin of this $t$-linear behavior of the heat current $\delta Q^i$ can be traced back to the corresponding $t$-linear behavior of the pressure $\delta p$ in Eq. (62).

In the absence of tilt in normal conductors, since the vector $\zeta$ is zero, there will be no preferred direction in the space, and therefore the vector transport coefficients $\gamma^i$ and $\beta^i$ remain inert. In tilted Dirac materials, these vector transport coefficients find a unique opportunity to become active and play a significant role by enabling a transport response to temporal variations of temperature.

Experimental applications of the above results might be interesting. We use graphene experimental data to estimate electric current for temporal gradients of tilted Dirac materials. Graphene has a high thermal conductivity which is usually estimated in the range $\kappa \sim 2000 - 4000 \frac{W}{mK}$ at room temperature [54]. Typically thermoelectric coefficients are determined in terms of Seebeck coefficient $S$ and thermoelectric figure of merit ($ZT$) which are more useful for exprimentalist:

$$ZT = \frac{\sigma S^2 T}{\kappa_e}, \tag{65}$$

$$\vec{J} = -\sigma S \vec{\partial} T, \tag{66}$$

$$\vec{Q} = -\kappa_e \vec{\partial} T, \tag{67}$$

where $\kappa_e$ can be measured using the Wiedemann-Franz Law $\frac{\kappa_e}{\sigma} = L_0 T$ where the Lorentz number $L_0$ is equal to $2.44 \times 10^{-8} W\Omega K^{-2}$. Seebeck coefficient for graphene can be measured as $S(T) = 100 \frac{T}{300[K]} \frac{\mu V}{K}$ [55] and electric conductivity is around $\sigma \sim 10^6 (\Omega.cm)^{-1}$. Based on the Kubo formula, the standard conductivity is given by current-current correlation funciton $\sigma^{ij} \propto \langle J^i J^j \rangle$, where $i, j$ denote the spatial directions. With the same reasoning, the vector transport coefficients is given by a new Kubo formula $\beta^i \propto \langle J^i J^0 \rangle$, where $J^0$ is the density rather than the current operator. On dimensional grounds, the latter correlation function is expected to be related to the conductivity $\sigma^{ij}$ by a natural velocity scale that can be nothing other than $v_F$. Therefore, at room temperature, we may approximate the effect of temporal gradients of temperature as follows:

$$\vec{J}_{\text{temporal}} \sim \frac{1}{v_F} \sigma S \partial_0 T \vec{\zeta}. \tag{68}$$

Typical temperature gradients in nanomaterials such as graphene are on the order of $\sim (1 - 20) \frac{K}{nm}$ [56–59]. The corresponding measured current densities are on the scale of $J^x \sim (10 - 200) \frac{\mu A}{nm^2}$. For a sample with 1nm × 1cm cross sectional area, the current is of the order of $10^9 \mu A$. Using a typical Fermi velocity for borophene $v_F \sim 10^5 m/s \sim 10^{14} nm/s$ renders the above spatial gradient to a temporal gradient of $10^{-14} Ks^{-1}$. This means that a temporal gradient of $\partial_t T \sim 10^{14} Ks^{-1}$ results in a current of $10^9 \mu A$ for a sample of 1cm width. Therefore with the available industrial heating rates of the order $\partial_t T \sim 0.4 \times 10^3 K/s$ [60] one expects currents on the scale of 4nA. Therefore the experimental demonstration of the vector transport coefficients requies industrial heating rates and samples of the $\sim$ 1cm width.

Generally, there can be materials with odd number of Dirac valleys [61], such as those on the surface of crystalline topological insulators [62]. But quite often, the Dirac cones come in pairs. This has to do with fermion doubling problem according to which, putting a Dirac theory on a (hypercubic) lattice leads to doubling of the number of Dirac fermions [63]. In the former case where the number of Dirac nodes is odd, there is no challenge and the vector transport coefficients are already active. But the latter case corresponding to lattices on which Fermion doubling occurs, requires some discussion: If the material is either inversion symmetric or time-reversal symmetric, requiring the invariance of the $ds^2 = -dt^2 + (d\vec{r} - \zeta dt)^2$ under $\vec{r} \to -\vec{r}$ or $t \to -t$ imposes a constraint on the tilt parameter of the two valleys: $\zeta_{\pm} = \pm\zeta$ [19]. When the number of Dirac valleys is even, and the effects in Eqs. (63) and (64) are odd functions of $\zeta$, the bulk currents from the two valleys cancel each other. Therefore, despite that the $\beta^i$ and $\gamma^i$ transport coefficients for a single valley are active, for the whole material hosting an even number of valleys they cancel each other's effect in an *infinite* system. However, if the materials lack either an inversion center or the time reversal symmetry (e.g. by placing it on a magnetic substrate) such that $\zeta_+ + \zeta_-$ becomes non-zero, then a net electric (heat) current enabled by the coefficient $\beta^i$ ($\gamma^i$) can flow in response to temporal gradients of temperature. Therefore, one possible rout to realization of the present effect is to search for a tilted Dirac cone in a material without inversion center.

Even if both time reversal and inversion symmetry are present, thereby leading to zero bulk currents, still the boundaries can be of help. The solution is based on the ideas of "valleytronics" [64] that are popular in planar (2D) materials. The essential idea is that imposing appropriate boundary conditions by cutting appropriate edges, one can create valley valves [65] at the other end of which the "populations" of the two valleys are imbalanced. In this approach, although the material is inversion and time-reversal symmetric and has even number of valleys with opposite tilt, $\pm\zeta$, that cancel out for *infinite* system, in finite systems with appropriately chosen boundaries, the population imbalance between the valleys gives rise to a net electric current $\sim \beta^i(n_+ - n_-)$ or heat current $\sim \gamma^i(n_+ - n_-)$ which is driven by the imbalanced population of the symmetric valleys. Such valley polarized effects resting on the boundaries, are expected to be important in mesoscopic systems.

## 4 Viscous flow of tilted Dirac fermions

So far we have brought up essential physics of TDFs for ideal fluid of electrons. It is now expedient to discuss the effect of viscosity in such systems. In this section, we focus only on electric response of the system and ignore the thermal perturbations. In this case, the energy and momentum conservation equations in Fourier space are respectively given as follows:

$$-i\omega\delta p = \frac{\gamma^2}{2+\zeta^2}\Big[-2i\omega(\epsilon_0+p_0)+\zeta^2\gamma\omega^2(\eta+\xi)\Big]\zeta_j\delta v^j + n_0\gamma\zeta_i E^i\,, \tag{69}$$

$$\frac{n_0 E^i}{\gamma} = \Big[\gamma\frac{\epsilon_0+p_0}{\tau_{\text{imp}}}\Big(1-i\omega\gamma\tau_{\text{imp}}\Big)\delta^i_j + 2i\omega\gamma^2\frac{\epsilon_0+p_0}{2+\zeta^2}\zeta^i\zeta_j$$

$$-\eta\omega^2\gamma^3\Big(\frac{4+3\zeta^2}{2+\zeta^2}\zeta^i\zeta_j + \zeta^2\delta^i_j\Big) - \xi_B\omega^2\gamma^3\frac{2}{2+\zeta^2}\zeta^i\zeta_j\Big]\delta v^j\,. \tag{70}$$

Solving for the perturbation in the velocity field, we find

$$\delta v^i = (\mathcal{F}^{-1})^i_j\Big[n_0 A^{jk}E^k\Big], \tag{71}$$

where $\mathcal{F}^{-1}$ is the inverse matrix of

$$\mathcal{F}^i_j = \mathcal{C}^i_j - \eta\omega^2\gamma^3\left[\frac{4+3\zeta^2}{2+\zeta^2}\zeta^i\zeta_j + \zeta^2\delta^i_j\right] - \xi_B\omega^2\gamma^3\frac{2}{2+\zeta^2}\zeta^i\zeta_j\,, \tag{72}$$

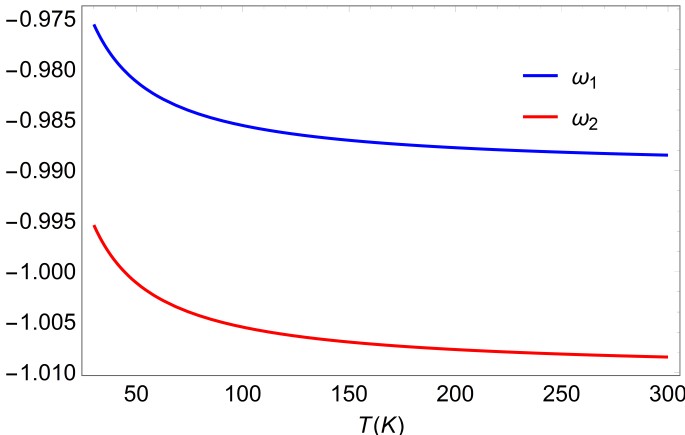

Figure 3: Temperature dependence of the poles of Eq. (73) in units of $i[\tau_{\text{imp}}]^{-1}$ for fixed $\zeta_x = \zeta_y = 0.1$

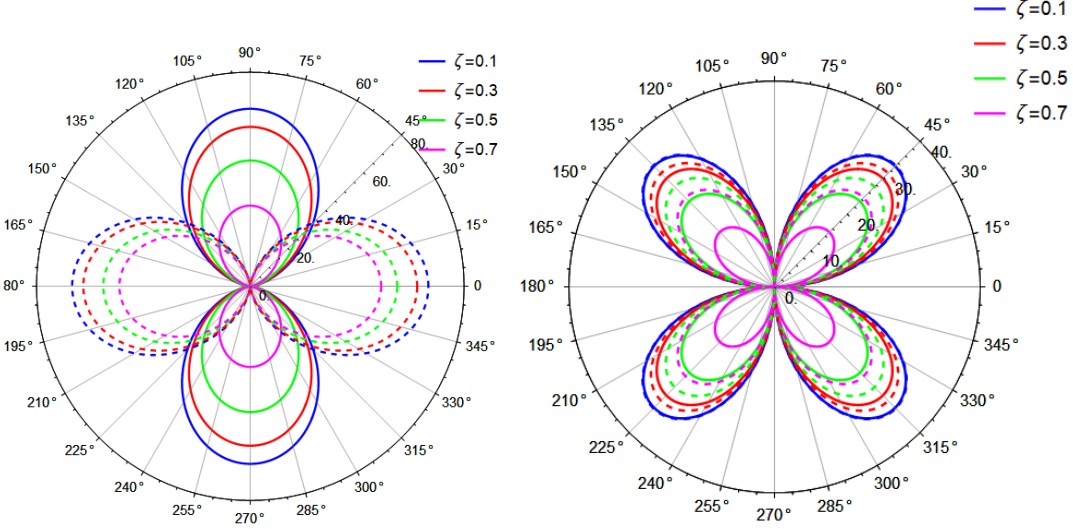

Figure 4: (Left) Polar dependence of the residue of $\sigma^{xx}$ for $\omega_1$ (solid lines) and $\omega_2$ (dashed lines) in units of $[\text{meV}.\tau_{\text{imp}}^2]^{-1}$ for viscous fluid at $T = 150$K. (Right) The same for $\sigma^{xy}$.

and $\mathcal{C}$ is the same as Eq. (39) for dissipationless fluid. The explicit form of inverse matrix is presented in the appendix. Then similar to the no-viscous fluid, by applying equations (12) and (71), we can read the electric conductivity coefficients in (9) as

$$\sigma^{ij} = \gamma n_0^2 (\mathcal{F}^{-1})^i_k A^{kj} + g^{ij} \gamma \sigma_Q. \tag{73}$$

This equation is quite similar to Eq. (40), except that the matrix $\mathcal{F}$ encompassing viscosity parameters replaces matrix $\mathcal{C}$ of non-viscose case.

The bulk viscosity at this order can be ignored [22] and shear viscosity can be controlled by temperature and its dependence for graphene is as follows [42]:

$$\eta = 0.45 \frac{(k_B T)^2}{\hbar (v_F \alpha)^2}, \tag{74}$$

where $\alpha = c\alpha_{\text{QED}}/(\varepsilon_r v_F)$ and $\varepsilon_r$ is the relative permeability of the material with respect to vacuum which for graphene can be estimated as $1 \leq \varepsilon_r \leq 5$ [22] and in this paper we have

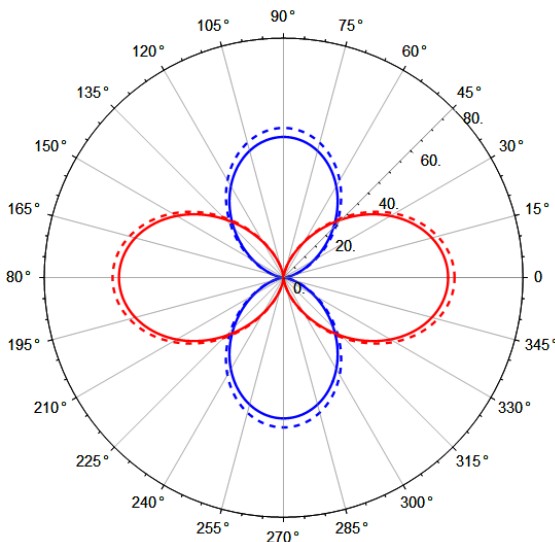

Figure 5: Comparison between the angular dependence of the residue of $\sigma^{xx}$ at fixed $\zeta = 0.5$ for viscous fluid (solid-lines) and ideal fluid (dashed lines). The poles $\omega_1$ and $\omega_2$ are shown by blue and red color. Spectral weights are larger for ideal fluid of TDFs.

assumed $\varepsilon_r = 3$ and $\alpha_{\text{QED}}$ is the fine structure constant. Essential scales are the temperature and the kinetic energy of the Dirac electrons set by $v_F$. Since $v_F$ in TDFs is comparable to graphene, we expect this formula to give a reasonable estimate of $\eta$ in TDFs.

As can be seen, the poles are given by zeros of the determinant of matrix $\mathcal{F}$ which is now a fourth degree polynomial and it must have two extra poles. It turns out that only two of these poles are causal and reduce to the $\omega_1$ and $\omega_2$ poles of the ideal fluid in Eq. (42) and (43). As can be seen in Fig. 3 the poles do not alter much by viscosity (which is controlled by temperature). In Fig. 4-Left we have given a polar plot of the residue of the $\sigma^{xx}$ at the $\omega_1$ (solid lines) and $\omega_2$ (dashed lines) poles. Similarly, in Fig. 4-Right we have plotted the same information for the Hall-like conductance $\sigma^{xy}$. As can be seen the two figures 4-Left and 4-Right parallel their ideal counterparts in Figs. 1 and 2, respectively.

As can be seen, their qualitative features are identical, so we do not expect the viscosity to heavily alter the conductivity of the TDFs. To see this in a more quantitative setting, in Fig. 5, we have shown a polar plot of the residues of the longitudinal conductivity $\sigma^{xx}$ for the viscous flow (solid line) and the ideal flow (dashed lines). As before, the vertical lobe (blue) corresponds to the Drude-like pole $\omega_1$, while the horizontal lobe (red) corresponds to the offspring (spacetime) pole $\omega_2$. Fig. 5 nicely shows how the viscous flow is continuously connected to the ideal flow. One further information that can be extracted from this figure is that the spectral intensity in ideal flow is larger than the viscous flow. Although the quantitative difference is small, but it indicates that the light can be better absorbed by the ideal TDFs than the viscous TDFs.

Now that we have given a comprehensive comparison between the conductivity tensor for the ideal and viscous flow of TDFs, we are ready to present the actual experimentally expected conductivity lineshapes. As noted in Eq. (42) and (43), and their viscous extensions in Fig. 3, both poles are purely imaginary and their real part is zero. This conforms to the intuition, as in a Fermi surface built on a tilted Dirac cone dispersion, one does not expect higher energy absorptions to take place. One still has the low-energy particle-hole excitations across the Fermi surface.

Fig. 6-Left shows the real (solid line) and imaginary (dashed line) parts of the longitudinal

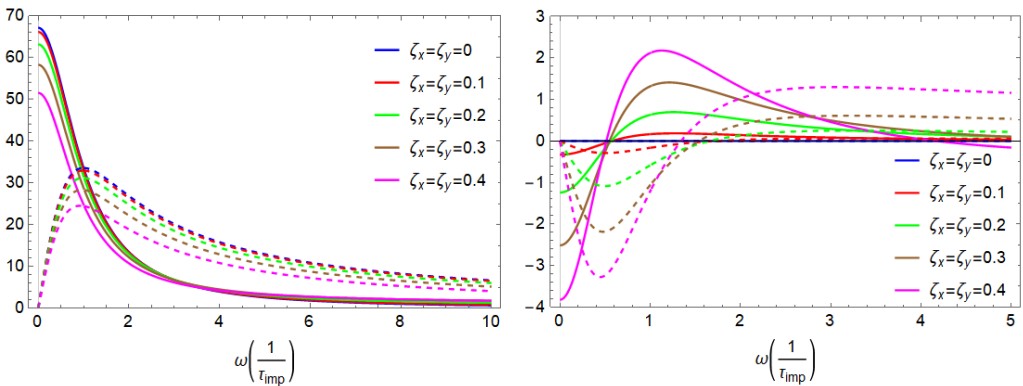

Figure 6: (Left) Real part (solid-lines) and imaginary part (dashed-lines) of $\sigma^{xx}$ for viscous fluid of TDFs at T=150 K in units of $[\text{meV}.\tau_{\text{imp}}]^{-1}$. The hidden structure inside the broad Drude peak can be revealed by polar dependence in Figs. 1 and 4-Left. (Right) The same for $\sigma^{xy}$.

conductivity $\sigma^{xx}$ for the fluid of TDFs. The viscosity corresponds to $T = 150K$. The polar angle is chosen to be $\theta = \pi/4$. The reason is that Fig. 1 and 4-Left suggest that the contribution of $\omega_1$ and $\omega_2$ poles becomes comparable along this direction. As can be seen, larger tilt values reduce the height of the Drude peak. Figure 6-Right shows the same information as Fig. 6-Left for the Hall-like conductivity $\sigma^{xy}$. Again we have plotted this curve for $\theta = \pi/4$ at which both poles have comparable contributions as demonstrated in Figs. 2 and 4-Right. The Hall-like conductivity in contrast to the longitudinal conductivity shows enhancement of the central peak upon increasing the tilt $\zeta$.

# 5 Summary and outlook

In this paper, we have investigated the hydrodynamics of tilted Dirac fermions that live in the tilted Dirac materials. The essential feature of these materials is that the "tilt" deformation of the Dirac (or Weyl if we are in three space dimensions) can be neatly encoded into the spacetime metric. As such, these materials are bestowed with an emergent spacetime geometry which distinguishes them from the rest of solid state systems. This metric encodes the long-distance structure of the complicated and rich content of the 8pmmn lattice. Putting it another way, the 8pmmn point group symmetry of the atomic scales becomes a metric at long wavelength scales [20, 21]. Therefore, as long the system has quasiparticles, even the strong electron-electron interactions responsible for the formation of the hydrodynamic regime are not able to destroy this metric structure if the lattice is not melted. What our hydrodynamic theory in this paper predicts is that the mixing of space and time coordinates in such solids gives rise to transport properties which have no analogue in other solid state systems where there is no mixing between space and time coordinates.

The first non-trivial consequence of the spacetime structure determined by tilt parameters $\zeta^i$ is that it gives rise to off-diagonal (Hall-like) transport coefficients *in the absence of magnetic field*. These Hall coefficients appear in both charge and heat transport. By generic symmetry structure of the spacetime in such solids, the conductivity tensors will be proportional to $g^{ij} = \delta^{ij} - \zeta^i \zeta^j$ that destroys the rotational invariance of the space. This is the root cause of *symmetric* Hall coefficient that does not require a magnetic field. The intuitive understanding of the anomalous symmetric Hall response $\sigma^{xy} \sim \zeta^1 \zeta^2$ come from the small tilt limit where the tilt can be viewed as a Lorentz boost parameter that converts part of the electric

field that is transverse to $\zeta$ into a magnetic field, whereby a Hall response can be generated. However, note that our theory is not limited to small tilts and is valid for arbitrary tilts.

The second non-trivial consequence of the structure of spacetime in these solids is the appearance of an additional $t$-linear contribution to the heat current for a constant temporal gradient of temperature $\partial_t T$ that is rooted in a corresponding $t$-linear behavior of the pressure in Eq. (62). In a finite system this leads to accumulation of charges that continues until the resulting electric field will cancel the pressure gradient. But when the system is part of an external circuit, the resulting heat current can be transmitted to the outside world for possible applications.

The third non-trivial and perhaps the most important aspect of transport in tilted Dirac materials which might have far reaching technological consequences is that a *temporal* gradient of temperature or electrochemical potential can be converted into electric or heat currents. The fact that the nature provides free $\partial_t T$ in hot deserts from mid-night to mid-day might transform this unique capability of TDFs into a technological concept. This property arises, because the structure of spacetime in tilted Dirac fermion solids is such that it allows for a mixing between space and time coordinates. The intuitive explanation of this effect is as follows: When the system is heated, i.e. $\partial_t T > 0$, in the absence of tilt, the entire heat goes into random motion of electron. A system with a non-zero tilt is related to the one with zero tilt by a Galilean boost. Boosting a the random kinetic motion of electrons will give rise to a net current. Therefore, it is not surprising that temporal gradients can drive currents, pretty much the same way spatial gradients can drive currents. We have introduced the notion of *vector* transport coefficients (as opposed to commonly used, *tensor* ones), to formalize and quantify these effects. The important technological advantage of such effects is that, via the vector transport coefficient $\beta^i$, a gradual heating of these materials ($\partial_t T$) generates electric currents.

Typical industrial heating rates are on the scale of $\partial_t T \sim 0.4 \times 10^3 \text{K/s}$ [60] that for samples of the width $\sim 1\text{cm}$, give total currents on the scale of 4nA that makes the vector transport coefficients assessible in the laboratory. Despite that it is possible to demonstrate this effect experimentally, in terms of the natural sources of the $\partial_t T$ in the hot desert, a typical temperature span of 40K during half cycle of earth rotation $\sim 40 \times 10^3$s will give $10^{-3}$K/s which is nearly 5 orders of magnitude smaller than the industrial heating rates. This means that large facilities of the order of $\sim 1\text{km}$ width are able to provide nA currents using the naturally available heating rates of the hot deserts.

Our theory presented in this paper is for a single tilted Dirac cone and is directly relevant to materials that host odd number of Dirac cones [61]. If the material at hand has both time reversal and inversion symmetry, there should be another Dirac node with opposite tilt that leads to the cancellation of the effects such as vector transport coefficients that are odd in $\zeta$. However, if the material lacks inversion symmetry, or time reversal (e.g. by proximity to magnetic substrate), or three dimensional Weyl materials that intrinsically break the time-reversal symmetry, the tilt parameters need not be opposite and still the effects survive. Even in the presence of both time-reversal and inversion symmetry, in mesoscopic devices, appropriately shaped boundaries can cause valley polarization that can imbalance the effects from two oppositely tilted Dirac cones [64, 65]. In this situation, the vector transport coefficients will be operating in an out-of-equilibrium setting. In fact, in such valleytronics setup, the valley polarization translates into an effective $\vec{\zeta}$ polarization which therefore activates a net non-zero vector transport coefficient.

The corresponding heat transport coefficient $\gamma^i$ makes materials with TDFs a suitable candidate in applications that require to guide the heat current into a given direction. This direction in TDFs is set by $\zeta$. Such a preferred direction is reminiscent of non-reciprocal spin currents in gradient materials [66]. The heating of a TDF material (corresponding to $\partial_t T > 0$) converts the heat to the work. This corresponds to the charging stage if this effect is employed

to construct a battery. The reverse situation, namely cooling with $\partial_t T < 0$ corresponds to the discharge stage of the battery where battery does work on the environment during which the heat is extracted from the system.

## Acknowledgments

S. A. J. thanks M. Mohajerani for providing an inspiring working environment during the COVID-19 outbreak. M. T. is supported by the research deputy of Sharif University of Technology. We appreciate helpful comments from the following colleagues: Reza Mansouri, Sohrab Rahvar, Sadamichi Maekawa, Farhad Shahbazi, Nima Khosrawi, Shant Baghram, Mehdi Kargarian, Abolhassan Vaezi, Saeed Abedinpour, Ali Esfandiar, Reza Ejtehadi, Takami Tohyama and Keivan Esfarjani.

**Funding information**   This project was supported by the Iran Science Elites Federation (ISEF) and the deputy of research, Sharif University of Technology, grant No. G960214.

## A   Appendix

In this appendix, we give explicit expressions for the transport coefficients of TDFs. The inverse matrix $\mathcal{F}^{-1}$ used in (72) is computed as follows

$$\mathcal{F}^{-1} = \frac{1}{f_0 + f_1\omega + f_2\omega^2 + f_3\omega^3 + f_4\omega^4} \begin{pmatrix} a_0 + a_1\omega + a_2\omega^2 & b_1\omega + b_2\omega^2 \\ b_1\omega + b_2\omega^2 & c_0 + c_1\omega + c_2\omega^2 \end{pmatrix}, \quad (A.1)$$

where the above coefficients are given by

$$\begin{aligned}
a_0 &= \gamma\frac{\epsilon_0 + p_0}{\tau_{\text{imp}}}, \\
a_1 &= i\gamma^2\frac{\epsilon_0 + p_0}{2 + \zeta^2}(\zeta_y^2 - \zeta_x^2 - 2), \\
a_2 &= 2\gamma^3\frac{\zeta_y^2}{2 + \zeta^2}\xi + \gamma^3\zeta^2\frac{(2 + \zeta^2)(1 - 2\zeta_y^2) - \zeta_y^2}{2 + \zeta^2}\eta, \\
b_1 &= -2i\gamma^2\frac{\epsilon_0 + p_0}{2 + \zeta^2}\zeta_x\zeta_y, \\
b_2 &= \gamma^3\frac{\zeta^2(2\zeta^2 + 5)\eta - 2\xi}{2 + \zeta^2}\zeta_x\zeta_y, \\
c_0 &= \gamma\frac{\epsilon_0 + p_0}{\tau_{\text{imp}}}, \\
c_1 &= i\gamma^2\frac{\epsilon_0 + p_0}{2 + \zeta^2}(\zeta_x^2 - \zeta_y^2 - 2), \\
c_2 &= 2\gamma^3\frac{\zeta_x^2}{2 + \zeta^2}\xi + \gamma^3\zeta^2\frac{(2 + \zeta^2)(1 - 2\zeta_x^2) - \zeta_x^2}{2 + \zeta^2}\eta,
\end{aligned} \quad (A.2)$$

and

$$\begin{aligned}
f_0 &= a_0 c_0, \quad f_1 = a_1 c_0 + a_0 c_1, \quad f_2 = a_2 c_0 + a_0 c_2 + a_1 c_1 - b_1^2, \\
f_3 &= a_2 c_1 + a_1 c_2 - 2b_1 b_2, \quad f_4 = a_2 c_2 - b_2^2.
\end{aligned} \quad (A.3)$$

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
