# Peer review of "Electron Currents from Gradual Heating in Tilted Dirac Cone Materials"

_SciPost Physics Core, doi:SciPost Phys. Core 6, 010 (2023)_

## Round 1 · Referee Report · Anonymous · 2021-11-19

Strengths

1. This paper presents a novel mechanism for generating electrical/heat currents via temporal temperature gradients by exploiting 2+1D tilted Dirac fermions in the hydrodynamic regime

Weaknesses

1. This paper has a large number of spelling and grammar errors.

2. This paper strongly relies on the preservation of an effective Lorentz theory, which is valid in the free electron picture. However, both impurity scattering and Coulomb interactions break this effective Lorentz symmetry, which they have not fully addressed. It's applicability to realistic condensed matter systems, which they have emphasized, is therefore questionable.

3. This paper does not give any numerical estimates for the strength or observability of generating currents via temporal temperature gradients, yet argues that such an effect is "easily observable."

4. One of their key results, an "accumulative heat transport," I believe is incorrect due to improper assumptions.

Report

The paper "Electron Currents from Gradual Heating in Tilted Dirac Materials" argues that for 2+1D tilted Dirac materials describable by hydrodynamics, they exhibit a novel behavior - purely temporal variations in temperature give rise to electrical and heat currents. Additionally, they give rise to an anomalous Hall response as well as "accumulative heat currents" - currents that depend on the exposure time of driving forces.

Their main idea - that a relativistic fluid with a non-diagonal metric gives rise to new transport coefficients coupled to dT/dt - is almost certainly correct. If this idea were properly developed in the paper, I would be more than happy to recommend the manuscript for publication. Unfortunately, it is not the case.

One undermining issue is the applicability of the relativisitic hydrodynamic description to physical systems, which is at the crux of their work. The effective Lorentz symmetry of is explicitly broken by Coulomb interactions as well as impurity scattering in physical systems, so it is not immediately obvious why the "curved spacetime" description should be valid in physical systems. The authors state that "strong electron-electron interactions are not able to destroy the emergent metric," arguing by analogy to the RG flow of the untilted Dirac fermion. However, for the tilted Dirac fermion, the fermi velocity v_F obtains an angular dependence; it is no longer protected under rotations and could run under RG differently in different directions, potentially destroying the spacetime metric.

Moreover, while this paper proposes the observability of these transport coefficients in experiment, it neither proposes an experimental measurement nor gives estimates for the observability of this effect. The only numerical estimates given in the entire paper are for dT/dt in various settings in Sec V, where they subsequently claim that "vector transport coefficients are easily observable effects in the laboratory." Without specific numerical estimates of a particular observable quantity, this statement seems unjustified.

I also believe the result on accumulative heat current in Sec 3.3 is incorrect. To obtain their result of Eq. 38, they assume (without justification) that the last term of Eq. 33 - delta P - is time independent. However, in Eq. 33 this implies that the LHS, namely delta P, is time-dependent. This contradiction invalidates their assumption, and thus invalidates the result.

Finally, while I agree that the transport properties are novel, it is not clear that "they have no analogue in other solid state systems where there is no mixing between space and time coordinates" as they write in Sec V. The key player is the angular-dependent Fermi velocity, which breaks both time-reversal and parity symmetries. The symmetry-breaking explains, for instance, the presence of a anomalous Hall effect. Similarly, the "vector transport coefficients" of Eq. 9 and Eq. 10 are symmetry-allowed without reference to an "emergent spacetime."

I would also comment that the quality of writing needs to be substantially improved, as there are numerous spelling and grammatical issues throughout the paper - in the last two sentences there are at least 4 issues!

Additional comments:
- The notation for bulk viscosity is not consistent. It shows up as zeta_B, xi_B, and xi. (see Eq. 8 and Eq. 13 and the surrounding text).
- Does the normalization |zeta| < 1 correspond to undertilted Dirac fermions?
- Why are chemical potential fluctuations not considered in Eq. 11?
- In Eq. 13, I believe the delta P should be a lower-case p, since the bulk viscosity appears at the very end.
- Eq. 17 is a momentum equation, but it is NOT the momentum conservation equation written in the text body just prior. An impurity scattering term has been introduced by hand.
- In Fig. 1, I believe the polar angle is defined as the angle between E and the tilt vector.
- In Fig. 3, the y-axis is unlabeled.
- In solving for the pole structure of oscillations, they look specifically for homogeneous flows, i.e. where no velocity gradients are created. However, this for instance excludes the hydrodynamic sound mode. In what limit are homogenous flows stable and dominant as opposed to the other modes in the system? In particular, I would at sufficiently high frequencies Fig. 7 and Fig. 8 to no longer be correct when spatial fluctuations become important.
- What is the numerical estimate for viscosity for Fig. 7 and Fig. 8? In particular, it requires an estimate for the relative permittivity constant (assuming you are using Eq. 45 to estimate this)
- "The metric encodes the long-distance structure of the complicated and rich content of the 8pmmn lattice. Putting it another way, the 8pmmn point group symmetry of the atomic scales becomes a metric at long wave-length scales." While I agree that the borophene lattice can give rise to tilted Dirac fermions near the Fermi surface and thus an effective metric, I do not think it is true that tilted Dirac fermions MUST come from an 8pmmn lattice.
- The appendix is missing.

In summary, I believe the manuscript requires major revision before acceptance into Scipost Physics.

  • validity: low
  • significance: ok
  • originality: good
  • clarity: ok
  • formatting: reasonable
  • grammar: below threshold

Author:  Seyed Akbar Jafari  on 2021-12-01  [id 1993]

(in reply to Report 1 on 2021-11-19)
Category:
answer to question
reply to objection

We appreciate the present referee for very careful reading of the paper and very insightful and instructive comments. The revised version of our paper will satisfy all the instructive comments of the referee.

We are happy that the referee agrees “almost certainly” that a non-diagonal metric gives new transport coefficient coupled to dT/dt. This is the essential point of our paper with far reaching technological significance.

The referee has correctly pointed out two microscopic mechanism as possible threat to our spacetime metric. We have made a recent progress in identifying the microscopic mechanism of the formation of tilt [a,b]. Our point of view based on the above research results is that the emergent space-time metric in certain quantum materials is the long-distance manifestation of the space group symmetry of the underlying lattice. This gives rise to a large degree of robustness to our metric description.

Therefore as long as strong interactions do not break those symmetries that support the metric (such as formation of density wave or nematic phases), our description of the anisotropy in terms of spacetime metric is expected to be valid. Aalbeit the interactions do renormalize the metric. So as long as, we are dealing with a conducting state (to which we are applying our hydrodynamic theory), the most serious harm that the interactions can cause to the metric is to renormalize the tilt parameters appearing in the metric.

Regarding referees concern stated as: “for the tilted Dirac fermion, the fermi velocity v_F obtains an angular dependence; it is no longer protected under rotations and could run under RG differently in different directions, potentially destroying the spacetime metric.” let us argue as follows: In a recent work [a] we have developed a microscopic understanding of the formation of the tilt for an example of 8Pmmn lattice. According to the picture developed in the above work, what referee is referring to as “anisotropy” is rooted in the difference between the second neighbor hoppings along vertical direction (see Fig. 2c of the above reference) and other second neighbor directions. Thinking in terms of this microscopic picture is very convenient. The Coulomb interaction is taking place on the same lattice. Given that the above difference in the hoppings is imposed on the electron system by underlying lattice, the Coulomb interaction is unlikely to be able to destroy it. As such, as long as the lattice is not molten, the above anisotropy continues to be present (although in a renormalized way).

The scattering from the impurities is also taking place on the same lattice structure that gives rise to our spacetime metric: As long as the concentration of impurities and the strength of the coupling to impurities is not strong enough to destroy the conducting state considered in our hydrodynamics approach (i.e. as long as electrons are not Anderson localized), the effects of impurity can be represented by a momentum relaxation time scale. This is what we have done in our theory.

Next let us remark on referees point of view that it is the “angular dependent Fermi velocity, which breaks both time-reversal and parity symmetries” that is responsible for the whole effect. We basically agree with this comment. However, the anisotropy can be of two type: (1) a generic anisotropy that is present even with upright Dirac cone (or even in systems without Dirac cone) and (2) the tilt-related anisotropy. The later part can be fully accommodated in a spacetime metric. Therefore in the later case, the consequences can be interpreted as properties of an “emergent spacetime”, while in the former case, such interpretation is not valid. In the revised version we will modify the formulation of the sentences to clarify this point and meet referee's concern.

Finally let us remark on referees concern about Eq. (38): "that the last term of Eq. 33 - delta P - is time independent. However, in Eq. 33 this implies that the LHS, namely delta P, is time-dependent. This contradiction invalidates their assumption, and thus invalidates the result." The last term is not $\delta P$, but rather $\partial_j \delta P$. So the RHS and LHS are two different functions and there is no contradiction. In fact Eq. (33) implies that the assumption of time-independent $\partial_j \delta P$ actually $leads$ to a t-linear $\delta P$. The consistency can be checked a-postoriori by taking the $\partial_j$ of the t-dependent $\delta P$ obtained in Eq. (33). If there are any concerns with how to derive the Eq. (33), itself, please see the attached pdf file (Note that to confirm with the uniform notation, we have changed all capital P to p).

As for the quality of writing, we will try our best as non-native speakers to improve it.

Response to Additional comments * “The notation for …”: Right. Thanks. Will be corrected. * “Does the normalization …”: Yes it does. * “Why are the chemical potential …”: We have assumed that the chemical potential is an independent thermodynamic variable besides the temperature. Being focused on the conductivity coefficients, for simplicity we have not considered the fluctuations of the chemical potential. Most likely including the chemical potential fluctuations will amount to replacing the electric potential with electro-chemical potential and the logic of the calculations will not change. * “In Fig. 13 …”: Yes, thanks for careful reading. * “Eq. 17 is a momentum…”: In the hydrodynamic framework, we usually treat momentum as an exactly conserved quantity. It is not true for electron fluid in metals and scattering of electrons off the impurities and phonons (mainly in high temperature) can not be neglected. The simplest way to modify momentum conservation equation is to introduce some term for impurities-electron interaction which is more important in low temperature. Based on the referee’s question, we clarify this point in more detail in the paper. * “In Fig. 1 …”: Yes, the reason is that the E is assumed to be along the x direction. * “In Fig. 3 …”: Thanks. It is modified in the revised version. * “In solving for …”: Thanks for pointing out this important point. The reason we have confined ourselves to zeroth order in gradient expansion is that we think the first order theory (accommodating the collective excitation) will have a much richer structure. In fact in [c,d] using Random Phase Approximation, we have found a bizarre "kink" in the collective excitation spectrum. It is an important question for us that requires a separate research work. The investigation of the stability of zeroth order theory can only be answered after having done the next order theory. * “What is the numerical estimate …”: For the case of 8Pmmn borophene, there is no data available on relative permittivity. But given that the Boron is the element just before Carbon, and that for many graphene samples the relative permittivity is in the 1-5 range, taking the average value of 3 seems to be reasonable that alpha=0.73 used in our paper to estimate $\eta$ in Eq. (45). * “The metric encodes …I do not think it is true that tilted Dirac fermions MUST come from an 8pmmn lattice”: Of course the referee is right. 8Pmmn lattice is one example of how a non-trivial space group in the long distance looks like a spacetime metric. In the revised version we reformulate the sentence to avoid this confusion.

[a] Y. Yekta, H. Hadipour, S. A. Jafari, arxiv:2108.08183 [b] A. Motavassal, S. A. Jafari, arxiv:2110.01906 [c] Z. Jalalimola, S. A. Jafari, Phys. Rev. B 98 (2018) 195415 [d] Z. Jalaimola, S. A. Jafari, Phys. Rev. B 98 (2018) 235430

Attachment:

hydroresponse.pdf

Anonymous on 2021-12-08  [id 2017]

(in reply to Seyed Akbar Jafari on 2021-12-01 [id 1993])
Category:
reply to objection

Regarding the robustness of the spacetime metric to strong interactions, the authors have pointed to previous works where they find tilted Dirac fermions in a non-interacting 8pmmn tight-binding lattice model (to next nearest neighbor order in hoppings). Effectively, they are arguing that so long as interactions simply renormalize these hoppings, one should preserves the metric behavior. This statement that the RG fixed point for strong interactions should be the non-interacting fixed point in the absence of lattice-breaking order parameters (i.e. CDW) is an unjustified assertion in my view. There's no particular reason this must be true; at strong interactions, quasiparticles may not even be well-defined objects. But even more simply, the metric that they write down is *not* protected under RG: the metric corresponds to a rigidly tilted cone, but as I argued previously, the RG may transform the $v_F$ differently in different directions, leading to a warped cone that cannot be described by a single tilt parameter and $v_F$.

Moreover, the validity of Eq. 33 is still suspect to me as it still seems self-inconsistent. Even if I assume $\partial_j \delta P$ is time-independent, as the authors have corrected me, the $\partial_j$ corresponds to a spatial derivative, so it will not remove any powers of $t$. The "equation" $\delta P \sim \int^t dt' \partial_j \delta P(t')$ still fails self-consistency under the given ansatz (unless trivially $\partial_j \delta P = 0$, where there is still no accumulative current).

I also agree with Ref. 3's complaint that they have not demonstrated that they have a Hall conductivity. What they have is an off-diagonal component of conductivity $\sigma_{xy}$, but to have a true Hall conductivity one needs the anti-symmetric piece of the diagonal term. That is to say, in the usual Hall effect one gets $\sigma_{xy} = -\sigma_{yx}$ (this is the meaning of $\epsilon_{ijk}\sigma_{jk} \neq 0$); without this, one can always find some basis to diagonalize $\sigma_{ij}$ since it would be a symmetric matrix. On symmetry grounds, due to the time-reversal breaking of the metric, I do expect a Hall conductivity to be present. However, it is not obvious to me whether or not this is present in Eq. 23, which is complicated to understand in its present form.

---

## Round 1 · Referee Report · Anonymous · 2021-11-23

Strengths

1. This work creatively pointed out novel phenomena that can be observed in the context of the tilted Dirac cone material.

Weaknesses

1. The author claimed to develop a hydrodynamic effective description of relativistic anisotropic fluid but did not considered the most general form of the stress-energy tensor and U(1) current
2. There is a very dubious key assumption in the construction namely the anisotropy represented by a vector $\zeta^\mu$ only enters the constitutive relation through the metric (as in Eq.(2)) with no anisotropic parameter in the ideal part of $T^{\mu\nu}$ and $J^\mu$. Furthermore, they also assume that the dissipative correction remains the same as in the isotropic case.
3. The construction of hydrodynamic for tilted Dirac cone material does not seems to be able to coupled to generic background metric and gauge field as the anisotropy is sourced by the metric alone.

Report

The work "Electron Currents from Gradual Heating in Tilted Dirac Cone Materials" by Moradpouri, Torabian and Jafari proposed a hydrodynamic descriptions of the tilted Dirac cone material. They then perform linear response analysis and discussed how various observable phenomena depends on the tilted parameters.

The hydrodynamic descriptions presented here is a class of relativistic anisotropic fluid which depends on external parameter $\zeta^\mu$. From this point of view a general form of the stress-energy tensor and $U(1)$ current is

$T^{\mu\nu} =T^{\mu\nu}[T,u^\mu,g_{\mu\nu},\zeta^\mu]\qquad \text{and}\qquad J^\mu =J^\mu[T,u^\mu,g_{\mu\nu},\zeta^\mu] $

This class of anisotropic fluid has been analyse in, at the very least, the quark-gluon plasma community albeit in 3+1 dimensions and without U(1) symmetry, see e.g. 1602.00573.

The construction of this manuscript however, did not consider the generic form of the stress-energy tensor pointed out above. Instead, they assume that the anisotropic property only enters through the metric as in Eq.(2). This assumption is motivated by earlier work [15,19] by the same authors that free fermion spectrum in the tilted Dirac cone material can be obtained from the untilted Dirac cone by a coordinate transformation that turns the flat metric into Eq.(2).

While this technique seems to help organising the computation in free theory, it is not clear (and unlikely) that, after the RG flow to hydrodynamic limit, equivalent to putting the relativistic hydrodynamic on the same metric. Moreover, an anisotropic fluid can still covariantly coupled to generic spacetime (see e.g. a method to use the background metric to source the temperature gradient in 0904.1975 and the a generic background to compute the retarded correlation function in 1205.5040 ) and not just putting the isotropic on a specific metric in Eq.(2).

Another way to say this is that, the logic of the derivation in this work is that

i) Putting the Dirac fermion in the metric (2) to mimic the spectrum of the tilted Dirac cone material
ii) Turn on interaction and flow to the hydrodynamic limit
iii) Obtain a certain anisotropic fluid

is equivalent to

i) Turn on interaction of the untilted Dirac cone material and flow to isotropic relativistic fluid
ii) Putting the isotropic relativistic fluid on the metric in Eq.(2) and claim that all anisotropic came from the coupling to the background metric

With such a dubious key assumption of the construction, it is very difficult to believe the validity and applicability of the remaining analysis in Section 3 and 4.

Requested changes

1. It would be great if there are is a solid argument that, for a tilted Dirac cone material, all the anisotropic effect do enters the constitutive relation in the way presented in this work. One honest way is to consider a generic anisotropic fluid and check that the additional ``transport coefficients" due to anisotropy beyond what presented in (4), (5), (8) do vanish in the system that the authors are interested.

2. If there is no such argument or there is an additional anisotropic transport, it would be great to reorganise the analysis in Section 3 and 4.

Doing so, however, will result in a very different paper.

  • validity: low
  • significance: good
  • originality: good
  • clarity: low
  • formatting: -
  • grammar: -

Author:  Seyed Akbar Jafari  on 2021-12-01  [id 1992]

(in reply to Report 2 on 2021-11-23)
Category:
answer to question

We thank the present referee for correctly capturing the essential points of our paper, and instructively raising insightful comments/criticisms.
As noticed by the referee, in this work we draw non-trivial conclusions associated with the presence of a non-trivial background metric in a solid-state material.
The main concern of the present referee is how to formalize the anisotropy in our hydrodynamic approach. The anisotropy in solids is not a new issue and comes from anisotropy in the ionic potential that is ultimately rooted in the lattice structure. The subject of our paper is to deal with a subclass of anisotropy that can be encoded into a metric.

The purpose of this paper is not to formulate the “most generic” theory of an anisotropic fluid. Even we do not choose to speak in terms of “anisotropy”. We have a clear logic for this: As the referee has correctly recognized, in the non-interacting limit, the tilt parameters $\zeta^i$ do not independently enter as anisotropy parameters, but are rather neatly encoded into a spacetime metric. At least in this limit, assuming that the energy-momentum tensor and the current depend separately on both metric AND tilt parameters is some sort of double counting that must be avoided. It is not clear how such a double counting goes away when the interactions become important.

We can however imagine a situation where referees suggestion of “most generic” theory of anisotropic fluid fits better: Goerbig and coworkers in [A] model a certain organic salt where even the upright (non-tilted) limit of the Dirac cone has a “genuine” anisotropy. This is the part that does not fit into the metric and must be considered separately. In this situation, the tilt part still goes into a metric, but the remaining anisotropy needs to be considered separately.

To debate the next important concern of the referee called by him/her as “dubious assumption” we need to think about a very essential question: What is the ORIGIN of the emergent spacetime metric? Form two of our recent works [B,C] we are led to view the emergent metric as the long-distance behavior of the space-group symmetry of the underlying lattice. Now the key “fact” (not assumption) is that, both the free theory, and the interacting theory (leading to hydrodynamic regime) are still mounted on the same mathematical lattice formed by the ion cores. Therefore, as long as the lattice is not molten, it is reasonable to assume that the same metric governs both free and interacting theory. Of course, interactions can and will renormalize the tilt parameters, because the tilt parameters are microscopically nothing but hopping along the second neighbor direction on a honeycomb lattice.

Referee's suggested literature: We will add a discussion corresponding to the following debate in our revised paper. * arxiv:1602.00573 : In this paper, the anisotropy is not included in the metric. The metric describes a simple Minkowski spacetime. But in our work, the entire tilt parameter is arranged into our metric. As a side remark, in Ref. [D] we tried to represent the polarization tensor of tilted Dirac cone materials by assuming that tilt parameters are “anisotropy” parameters. But the end result was a nice little formula (satisfying Ward identity) where the entire effect of tilt was compactly encoded in our metric. It seems that as Anton Zee in his QFT in a Nutshell book puts, “metric comes looking for us”. Therefore, we are led to view the tilt parameters as entries of some metric, rather than generic anisotropy parameters. From materials and condensed matter point of view, this point of view is very important, because it opens up a vast playground for synthesis of “geometric forces” in solids. * arxiv:0904.1975 : In this work, a space dependent metric can source a temperature gradient in their Eq. 18. But in our work, it is important to notice that our solid-state metric is not a dynamical metric (as in gravity). * arxiv:1205.5040: As pointed out, the purpose of our work is to deal with a type of anisotropy that can be encoded into a (non-dynamical) spacetime metric. Otherwise the generic anisotropy is well studied subject in the solid state physics. Our aim is to draw conclusions that can be attributed to an underlying metric. Of course we agree that the retarded response functions can be obtained by perturbing the metric. But the full-fledged machinery of the above beautiful lectures of Kuvton is not needed in our treatment.

Requested changes: The referee has given us two options: (1) To present an argument defending why in the hydrodynamic regime still the tilt parameter can be encoded into a metric. (2) If not, to reorganize sections 3, 4. 
 Given our two recent works [B,C] suggesting that the emergent “solid-state spacetime” actually represents the long-distance behavior of microscopic symmetries of the lattice (i.e. the space group), as long as the lattice is not molten, the same metric will continue to encode the effect of the tilt. We hope that the referee will accept this as “solid argument”.

References: [A] M. O. Goerbig et al, Euro. Phys. Lett. 85 (2009) 57005 [B] Y. Yekta, H. Hadipour, S. A. Jafari, arxiv:2108.08183 [C] A. Motavassal, S. A. Jafari, arxiv:2110.01906 [D] Z. Jalalimola and S. A. Jafari, Phys. Rev. B 100 (2019) 075113

---

## Round 1 · Referee Report · Anonymous · 2021-11-28

Strengths

I thought the authors tried to study a very interesting system and it is a problem worth studying.

Weaknesses

The paper seems to have major and qualitative errors in the analysis, to the point where I do not believe the results in present form. I think the paper needs to either be withdrawn or completely re-written with new and corrected calculations, unfortunately.

Report

Overall I do not think this paper should be published in SciPost Physics -- certainly not in present form, and if revised, only if the revisions are sufficiently substantial that they almost lead to a complete overhaul of the paper. There are a number of key issues, many pointed out by other referees already.

1) I agree with other referees that the presence of Coulomb interactions etc. will in general further break the symmetry from Eq. 1. Of course that does not mean the problem is uninteresting or not worth publishing, but the comments above/below Eq. (1) are too strong and should be fixed: interactions could in principle qualitatively modify things.

2) The authors seem to use zeta_B for bulk viscosity, but then switch to xi -- please make uniform!

3) I assume the 2 poles omega_{1,2} just amount to effectively different relaxation times for momentum parallel to or perpendicular to. \zeta, but this whole discussion is likely wrong: see the next 2 points.

4) The authors' comment that they get a Hall conductivity are wrong. Hall conductivity in 3 dimensions would come from a non-vanishing epsilon_{ijk}sigma_{jk}, which they do not have. It is just that if zeta is chosen to not align with a coordinate axis (but, why make that choice here given otherwise isotropic medium...) then there are off diagonal terms in sigma_ij.

5) I am extremely skeptical of the claim around Eq. (38) that the authors can get a heat current that grows linearly with time. The issue appears to essentially be that because of zeta, the pressure gradient term cannot be balanced in Eq. (19). However, I think that most likely within the authors' current framework there would need to be some sort of momentum-relaxation type term added into Eq. (19) related to zeta. Otherwise all the linear response transport calculations are going to be ill-posed because Eq. (19) could not possibly be satisfied. Yet such divergences did not seem to show up in the discussion around Eqs (22-28), so I also doubt those calculations are correct as written.

Honestly, I didn't really read Section 4 because I think there were so many mistakes in the analysis of Section 3 that it would not be worthwhile.

If the authors want to resubmit this paper, I think they need to extremely carefully revisit the starting point and assumptions in the model, etc. I would honestly advise that for a system like this it might be useful to start by trying to study a more microscopic Boltzmann model for transport, where their starting points like Eq. (19) can be more carefully checked.

Requested changes

See report.

  • validity: poor
  • significance: good
  • originality: good
  • clarity: ok
  • formatting: acceptable
  • grammar: acceptable

Author:  Seyed Akbar Jafari  on 2021-12-02  [id 2003]

(in reply to Report 3 on 2021-11-28)
Category:
correction

In the second paragraph of our response to item 4 of referee 3, we need to replace:
"The only way this expression can be zero is that ..." with "The only way this expression can become NON-zero is that ..."

Author:  Seyed Akbar Jafari  on 2021-12-02  [id 2001]

(in reply to Report 3 on 2021-11-28)
Category:
answer to question
reply to objection

We thank the referee for taking time to read our paper up to section 3. Referee's comments appear in Italic font.

REFEREE: I thought the authors tried to study a very interesting system and it is a problem worth studying. Response: We thank the referee for appreciating that the study of tilted Dirac cone materials in terms of spacetime metric is interesting.

REFEREE: The paper seems to have major and qualitative errors in the analysis .... Response: We hope that our well referenced response in the following will clear the misunderstandings and will convince the referee that our work deserves a better description.

REPORT: 1- I agree with other referees that the presence of Coulomb interactions etc... Response: The same concern has been raised by other referees. As we have pointed out in our response to referees 1 and 2, the root cause of the appearance of spacetime metric is the transmutation of the short-distance space group into spacetime metric in the long-distance. We would like to further point out that, our starting pint is NOT a microscopic Hamiltonian containing Coulomb forces. Rather, based on the logic that the spacetime metric is a long-distance manifestation of the underlying lattice, and assuming that the lattice structure does not change as we crank up the interactions from zero to the hydrodynamics limit, we formulate our hydrodynamic theory in a fixed (non-dynamical) background metric. As side remarks, in referees formulation of the symmetry breaking: (i) We do not see what is the meaning of "further" in "further break the symmetry from Eq. (1)". Does he/she mean anything more than the group of isometries of our metric? Does he/she refer to a possible order parameter? (ii) In referees expression "comments above/below Eq. (1) are too strong and should be fixed" the referee has not precisely referenced which statement below and above Eq. (1) is meant and how far below or above Eq. (1) is meant. We would be happy to address his/her concerns once he/she sharply specifies what is too strong.

2- The authors seem to use zeta_B for bulk viscosity, but then switch to xi -- please make uniform! Response: Sure. Thanks. We have already taken notice of this typo based on the report of referee 1.

3-I assume the 2 poles omega_{1,2} just amount to effectively different relaxation times for momentum parallel to or perpendicular to. \zeta, but this whole discussion is likely wrong: see the next 2 points. Response: In this comment, the referee is referring to next two points. What we find in point 5 is that by stating "there would need to be some sort of momentum-relaxation type term added into Eq. (19) related to zeta" the referee is requiring us to add a "momentum-relaxation type term". If the referee thinks that the "momentum-relaxation type term" has not been included, then how can the $\omega_{1,2}$ poles obtained in our theory "just amount to effectively different relaxation times for momentum parallel to or perpendicular to. \zeta"?

4- The authors' comment that they get a Hall conductivity ... Response: Others have also obtained a Hall conductivity that agrees with ours. For example in Ref. [I], in their Eq. (25) they find a Hall conductivity proportional to $\sin\alpha\cos\alpha$ where $\alpha$ is the tilt angle which in our language will be proportional to $\zeta^1\zeta^2$ agreeing with our result. It is nice that we obtain similar result using our metric. Hence we come up with a new interpretation, that the non-zero Hall conductivity in this case is a property of the underlying spacetime. Please note that the coefficient of the $\sin\alpha\cos\alpha$ in the above reference vanishes when the tilt parameter goes to zero. Our metric-based insight into such a non-zero Hall coefficient provides the following intuition into the origin of such Hall effect: The transformation from the Minkowski metric $ds^2=-dt^2+(d\vec r)^2$ to the tilted Dirac materials metric $ds^2=-dt^2+(d\vec r-\vec\zeta dt)^2$ involves a Galilean transformation which in the small $\zeta$ limit can be viewed as a Lorentz boost. It is this Lorentz boost that folds parts of the in plane electric field $\vec E$ into an "effective" magnetic field $\vec b_\zeta\propto \vec\zeta\times \vec E$. This is the origin of non-zero Hall coefficient. We hope that this simple and intuitive argument will convince the referee that our Hall coefficient not only is not wrong, but is a novel property of the underlying spacetime. When $\zeta$ is not small, the complete isometries of our metric is worked out in our earlier works [III]. The end result is that, the emergent spacetime of tilted Dirac cone materials is such that, if we are to interpret the results in terms of our Galilean solid-state intuition, we have view the tilt parameters $\vec\zeta$ appearing in the metric as entities that transfigure among the other things, $\vec E$ and $\vec B$ to each other [III].

We find the following comment difficult to comprehend about which we would like to debate: “Hall conductivity in 3 dimensions would come from a non-vanishing $\epsilon_{ijk}\sigma_{jk}$, which they do not have.”. response:This expression seems to be a contraction of Levi-Civita symbol $\epsilon^{ijk}$ with the conductivity tensor $\sigma_{jk}$ (where $j,k=1,2$ for spatial indices). The only way this expression can be zero is that (a) either the $i=3$ or (b) $i=0$. The case (a) is related to a problem in 3 space dimensions and is not the focus of our paper. The option (b), relates the Hall conductivity $\sigma_{xy}$ to zero'th component of some vector, say $J^0$ which is again not clear what it means. Option (c) is that perhaps the referee means the Chern-Simons term (effective field theory of quantum Hall problem). Then quoting from chapter VI.2 of Anton Zee's book on QFT [II] the $\epsilon^{\mu\nu\lambda}$ tensor will be participate in the Lagrangian as $\frac{k}{4\pi}\epsilon^{\mu\nu\lambda}a_{\mu}\partial_\nu a_\lambda$ where the coupling $k$ is the off-diagonal Hall conductivity, $\sigma_{xy}=\sigma^H$. Again this will be irrelevant to our problem, as the Hall conductivity $\sigma^{ij}\propto \zeta^i\zeta^j$ in our work is classical (non-quantized) Hall coefficient. If the referee has any other thing than the cases (a-c) in mind, we cordially request him/her to sharpen this statement in order to continue the debate.

5- I am extremely skeptical of the claim around Eq. (38) ... Response: If your skepticism concerns the algebraic steps leading to Eq. (38), we refer you to the attached file. Regarding “The issue appears to essentially be that because of zeta, the pressure gradient term cannot be balanced in Eq. (19).” we somehow agree that that $\vec\zeta$ being a property of the emergent spacetime, operates in the hydrodynamic flow as some sort of constraint that tends to guide the stochastic motions. So it is a property of such emergent spacetime and balancing it would eliminate the whole effect of the spacetime. In response to “However, I think that most likely within the authors' current framework there would need to be some sort of momentum-relaxation type term added into Eq. (19) related to zeta.” we would agree in the following sense: The momentum relaxation term in momentum Eq. (20) (NOT eq. 19) prevents the conductivity from divergence. Likewise one expects that a similar energy-relaxation term (due to the heat transferred from electron system to the lattice) needs to be added. Corresponding to this debate, we will add discussions into the paper. We thank the referee for raising this debate. In reaction to (iii) “Yet such divergences did not seem to show up in the discussion around Eqs (22-28), so I also doubt those calculations are correct as written.”, we would agree with the referee and can explain it as follows: Eqs. (22-28) express the response of electric current to external stimuli, where already existing momentum relaxation in our theory prevents it from diverging, while the divergence in the heat current is because the constant exposure to heat by a positive $\partial_t T$ would lead to more and more heat transport in the system. I hope the referee is convinced that Eq. (38) is correct result for a theory of electron fluids. Noting that through the cycle of Earth around the itself, $\partial_t T$ has to be a periodic function, and not a uniformly increasing function can be a remedy of the divergence in practical devices. In the revised version we will include discussions related to this debate.

Last comment: Honestly, I didn't really read Section 4 because I think there were so many mistakes in the analysis of Section 3 that it would not be worthwhile. Response: We thank you for reading 3 sections of our work. Peer review is a voluntary action. Even if our work is not worth reading, repeated use of despising expressions is not necessary. The power of reason is enough to refute our arguments.

References: [I] Y. Suzumura, et al, J. Phys. Soc. Jpn., 83 (2014) 023701 [II] A. Zee, Quantum Field Theory in a Nutshell, Princeton Univ. Press (2010) [III] S. A. Jafari, Phys. Rev. B 100 (2019) 045144.

Attachment:

hydroresponse.pdf

---

## Round 2 · Referee Report · Anonymous · 2022-2-28

Strengths

1. They detail a novel mechanism to generate electrical/thermal currents via temporal fluctuations of temperature by exploiting tilted Dirac fermions in 2+1D.

Weaknesses

1. While they argue for experimental realizability, it is not yet fleshed out in this paper.

Report

This is my second pass of this paper, and in its current state I believe it meets the SciPost criteria for publication.

As before, this paper primarily outlines a novel procedure to generate electrical/thermal currents from temporal fluctuations of temperature by utilizing tilted Dirac fermions in 2+1D; the tilt provides an effective metric which mixes spatial and temporal coordinates, realizing this effect.

Most of the previous issues have been sufficiently addressed or corrected in their updated paper. However, I still find the discussion on experimental detection lacking. While in their submission form, they state that "the vector transport coefficients ... will be comparable to thermoelectric coefficients," I still do not see numerical estimates of this in the paper. What *is* present are estimates of temperature gradients, which are (in theory) controlled by the experimenter. In other words, for a given temperature gradient/temperature fluctuation, how much current do I expect to measure in these tilted Dirac fermion systems? Without such estimates, I find claims of experimental feasibility hard to substantiate.

  • validity: good
  • significance: ok
  • originality: good
  • clarity: good
  • formatting: reasonable
  • grammar: reasonable

Author:  Seyed Akbar Jafari  on 2022-05-17  [id 2486]

(in reply to Report 1 on 2022-02-28)
Category:
answer to question

We thank the present referee for his/her positive evaluation of the current version of our paper. We have revised the text by adding texts around Equations (65)-(68). Estimates are based on the thermal conductivity measurements performed on graphene samples.

The actual numbers required by the referee are as follows: For industrially available heating rates on the sale of 1000 K/s, samples of the width ~1cm are expected to give ~4nA currents. Larger samples can provide larger currents. Details are given in the revised text.

---

## Round 2 · Referee Report · Anonymous · 2022-3-3

Strengths

The current version further elaborate on the key assumptions on the applicability of hydrodynamics with nontrivial metric in material with tilted Dirac cone. This resolve the main concern raised in the previous version.

Moreover, the computational details are now much more readable. A few strange terminologies (e.g. "Hall" conductivity in the previous version) and notations inconsistencies are removed.

Overall, this is a much better manuscript than in the previous version.

Weaknesses

I think there are still a few typos and some terminology which are not very precise. These are minor details but should be properly fixed.

Report

I think that the assumptions stated in the current version by the authors are reasonable and, as already stated in the previous report, that the phenomena pointed out in the manuscript are interesting.

Requested changes

There are still minor wordings and typos that I found, for example
- In the last two sentence of the abstract, I'm not sure what the term "electric energy" means. Perhaps, the authors would like to use a more precise term such as electric current.
- On paragraph below equation (1), the hyperlink is broken i.e. " are fermionic or bosonic [?, 20]"
- On the second paragraph of Section 2, "The later is valid for the emergent spacetime" where it should be "The latter..."

  • validity: ok
  • significance: good
  • originality: good
  • clarity: good
  • formatting: reasonable
  • grammar: reasonable

Author:  Seyed Akbar Jafari  on 2022-05-17  [id 2487]

(in reply to Report 2 on 2022-03-03)

We thank the present referee for his/her positive evaluation of the current version of our paper. In the revised version
(1) We have replaced the "electric energy" with "electric current".
(2) We have fixed broken links to references and have updated some other references.
(3) We have fixed other typos.

We hope that the current version will satisfy the present referee.

---

## Round 2 · Referee Report · Anonymous · 2022-3-6

Strengths

The paper is on an interesting topic, as I wrote previously.

Weaknesses

I am concerned that the paper still has fundamental mistakes in it. It seems as though the response of the system might be incompatible with basic principles, and I believe the authors still need to address some important points listed in my report.

Report

1) Normally in hydrodynamics if you turn on an "external temperature gradient" as the authors did in Section 3.1.1, you could "cancel" the effect by having the actual temperature change so as to cancel the effect out; similarly, the electric field could in principle be cancelled off by a x-dependent chemical potential. For example taking the latter case (in the absence of tilt) I'd have

d_mu T^mu i = n E^i = d_i P = n d_i mu

so taking mu = E^i x_i I could find consistency. It seems this cannot happen in Eq. (27) because there would be a zeta^j d_j mu term on the left hand side of the equation but there is no similar term on the right hand side. This makes me worry that the calculation in Section 3 might in some way be inconsistent. It seems like a plausible argument on physical grounds that you wouldn't want the external electromagnetic fields to see the metric tilt...but that may have then led to this later issue, and I worry this is a serious enough point that it might mean this plausible assumption was wrong. I am not 100% sure if this is a mistake that needs to be fixed or just a funny feature of this system; however, at a minimum some discussion of this point should be had, and I do honestly lean towards thinking that the calculation is just not correct in some way.

2) In Eq. (33), why is there a tau_imp that appeared seemingly out of nowhere? I.e. it has nothing to do with d_t T, but was not included in Eq. (27).

3) In Section 3.2 I am still concerned about a few points. If I use the memory matrix formalism (see e.g. the discussion in "Holographic Quantum Matter"), I would expect that if I have long-lived momentum that the conductivity (ignoring incoherent part for a moment) could be approximated as

\sigma_{ij} = n^2 \Gamma_{ij}^{-1},

where \Gamma_{ij} relates to momentum relaxation and would encode the anisotropy, while n = \chi_{JP} is a susceptibility which is fixed. Even in an anisotropic system, \chi_{JP} does not become anisotropic in general. In this formalism it seems as if the structure that arises is rather different though. Since the derivation based on memory matrix methods would be rather formal and microscopics-independent, it should be valid for this system too. So the authors either ought to explain why the above expectation is wrong, or correct Section 3.2 (and possibly earlier sections too) in order to resolve the issue.

4) Below Eq. (43), omega_2 can be arbitrarily large compared to omega_1. More importantly, I would expect that you would find in this anisotropic system that (if we align the tilt axis with a particular direction, let's say x) that sigma_{xx} has a Drude peak with a different pole than sigma_{yy} and sigma_{zz}. If there are 2 poles visible in the same 'component' of sigma, it would simply be that the axes were not aligned but one could always choose 'smart' axes where sigma was diagonal and the pole structure was clearly separate. Is this what happens in this theory? Why or why not?

5) I strongly object to calling this "Hall response" or "Hall-like response", it is simply anisotropic conductivity which is well-established in anisotropic materials. I also don't know why this response would be considered "anomalous" in a tilted Dirac material which is anisotropic?

6) I believe the argument at the top of page 12 is almost certainly wrong: if there were an emergent B-field that was giving rise to Hall-like transport, then the conductivity would become antisymmetric?

7) I worry the fact that the system becomes heated uniformly in a uniform temperature gradient in Eq. (54) might be a consequence of an incorrect implementation of temperature gradient in the earlier section 3.1, as per my previous point above. Typically *within linear response* one can always find steady-state solutions to the equations, and that seems not to be true here, which is a bit concerning!

Requested changes

Please address the points in the report.

  • validity: ok
  • significance: good
  • originality: high
  • clarity: low
  • formatting: acceptable
  • grammar: acceptable

Author:  Seyed Akbar Jafari  on 2022-05-17  [id 2488]

(in reply to Report 3 on 2022-03-06)
Category:
answer to question

We thank the present referee for insisting on the points that leads to improvement of our paper. Please see the attached file for a detailed response to your comments/criticisms.

Attachment:

response2nd.pdf

---

## Round 2 · Author Response

We are happy that the scrutiny of the referees and editors has led to a reformulation of the most important results of our paper and a better understanding of the novel results of our paper. Here we give a response to essential criticisms of the referees.

  1. One major concern shared by the referees is that effect of Coulomb interactions on the structure of the spacetime. Referees are right. As long as the electron fluid is still described by quasiparticles, the metric structure is preserved, although the tilt parameters can be renormalized with respect to those obtained from band structure. We have clearly stated this point in the introduction of the paper.

  2. Another major concern of the referees is the nature of Hall response. In the revised version we have argued that the rotational invariance leads to the property that the fact that the off-diagonal (Hall) response in normal conductors has only antisymmetric part. Any symmetric part would vanish by a rotation (that leaves the system invariant). However, in the tilted Dirac cone materials, the anisotropy arising from the tilt breaks the rotational invariance and hence the Hall response can not be used to rule out symmetric Hall response. Of course such a Hall-like response is anomalous in the sense that it does not require external B field. Upon application of B field, the antisymmetric part of the response also becomes active.

  3. Regarding the controversial Eq. (38) of the previous version of the paper, we have now added a new section 3.1 that properly takes "external" sources into account. Furthermore, the interpretation as "accumulative" current applies only to finite samples. We have argued that the t-linear pressure gradient will be eventually balanced by a counter reacting electric field. When the system is part of an external circuit, such heat/electric currents flow in the circuit and do not lead to accumulation. We have corrected the earlier misinterpretation and have clearly stated it in the paper. Furthermore, proper handling of the external sources resolves all the issues about the self-consistency of a t-linear term. In the revised version where the sources are properly included, the t-linear pressure that is the root cause of all t-linear terms is actually rooted in the t-linear increase of the temperature (external source) for which we have explicitly solved the equations.

  4. Regarding the concern of referee 2, we have clearly stated and separated the effects that arise from "generic" anisotropy from the particular form of anisotropy that can be encoded into the spacetime metric. So our line of thought is to study only the forms of anisotropy that can be encoded into a spacetime metric. Because then the resulting effect can be attributed to the structure of the underlying spacetime.

  5. We have tried our best to carefully rewrite the paper in order to reduce the language/grammatical errors.

  6. As for numerical estimates, in the conclusion we have given estimates based on the state of the art cooling rates Technologies. In this case, the vector transport coefficients introduced by us, after conversion to spatial gradients via the Fermi velocity, will be comparable to thermoelectric coefficients and hence they can be in principle demonstrated in the laboratory.

---

## Round 2 · List of Changes

In the revised version we have highlighted the major changes with red color to assist referees and the editors in clearly seeing the changes that are summarized below:
- Inconsistencies in the notations are resolved.
- The entire abstract, introduction and conclusion was rewritten.
- An entirely new subsection 3.1 has been added that properly deals with "external sources". The lack of such section was source of our own confusion about the self-consistency of t-linear currents.
- A discussion in section 3.3 is added to argue why a symmetric Hall response can arise in our system. To emphasize this, every "Hall" in the paper has been replaced by "Hall-like".
- We have totally rewritten the discussion of t-linear currents in section 3.4 and have corrected our misinterpretation of "accumulative currents" that only applies to finite (closed) system.
- After properly adding the external temperature sources to the theory, the vector transport coefficients in Eqs. (59) and (60) of section 3.5 are also corrected. These are essential result of our paper and quantify the response of the fluid to gradual heating. This section is also entirely rewritten.
- The section 4 on viscous fluids has not changed much except for small rephrasing here and there.
- The section 5 on conclusion has also been rewritten to account for the above list of changes.

---

## Round 3 · Referee Report · Anonymous (Referee 3) · 2022-7-7

Report
I think I mentioned in an earlier report that other methods (whether memory matrix or kinetic theory) would be really valuable here, because I believe they would give alternative perspectives on whether the phenomena the authors are describing here are genuine or due to subtle mistakes. For example one would be able to determine whether the physics discussed around Eq. 54 is due to something pathological about boundary conditions, the analysis, or might really be physical.
I personally would strongly hesitate towards publishing this paper, both for the reason I mentioned above, but also because there were some earlier points I had made that I think the authors did not fully take seriously (e.g. "Hall-like" response) and I am concerned this demonstrates some fundamental misunderstandings about the physics which may have affected the remaining analysis. Anyway, while I doubt this paper is all correct, it still calls attention to an interesting problem. I'll let the editor make the final call, but I don't think I have much more to add or to help with.

Seyed Akbar Jafari on 2022-07-13 [id 2658]
We are happy that the present referee's grading of the version 3 suggests "High" for the originality, "Good" for the clarity and significance, and suggests "Ok" for the validity of our results.
Please note that within Eqs. (54)-(58) we have found two sets of solutions: Those with zero pressure gradient and those with non-zero pressure gradient.
As for the "Hall-like" response, we have argued that since the ordinary rotations are not symmetries of the new spacetime, the symmetric part of the response can not be "rotated away".
We thank the present referee for the improvements arising from his/her insightful comments.

---

## Round 3 · Referee Report · Carlo Beenakker (Referee 4) · 2022-8-26

Strengths
Weaknesses
Report

---

## Round 3 · Author Response

List of changes
1- We have added discussions after Eq. (54) to show that steady-state solutions are also possible. But in addition to steady- state solutions, there are also non-steady-state solutions that are of interest to us. Hopefully this will convince the referee 1 that our theory does not miss the steady-state solutions. 2- Around newly added equations (65)-(68) we have added estimates based on the measurements done on graphene samples to estimate that for tilted Dirac cone samples that are ∼ 1cm wide, currents of ∼ 4nA are attainable. 3- We have corrected typos here and there.

---

## Round 3 · List of Changes

1- We have added discussions after Eq. (54) to show that steady-state solutions are also possible. But in addition to steady- state solutions, there are also non-steady-state solutions that are of interest to us. Hopefully this will convince the referee 1 that our theory does not miss the steady-state solutions. 2- Around newly added equations (65)-(68) we have added estimates based on the measurements done on graphene samples to estimate that for tilted Dirac cone samples that are ∼ 1cm wide, currents of ∼ 4nA are attainable. 3- We have corrected typos here and there.

---

## Editorial Decision

published